# Umbilical cord blood-derived ILC1-like cells constitute a novel precursor for mature KIR⁺NKG2A⁻ NK cells

**Sabrina Bianca Bennstein[1], Sandra Weinhold[1], Angela Riccarda Manser[1], Nadine Scherenschlich[1], Angela Noll[2], Katharina Raba[1], Gesine Kögler[1], Lutz Walter[2], Markus Uhrberg[1]\***

[1]Institute for Transplantation Diagnostics and Cell Therapeutics, Medical Faculty, Heinrich-Heine University Düsseldorf, Düsseldorf, Germany; [2]Primate Genetics Laboratory, German Primate Center, Leibnitz-Institute for Primate Research, Göttingen, Germany

**Abstract** Despite their identification several years ago, molecular identity and developmental relation between human ILC1 and NK cells, comprising group 1 ILCs, is still elusive. To unravel their connection, thorough transcriptional, epigenetic, and functional characterization was performed from umbilical cord blood (CB). Unexpectedly, ILC1-like cells lacked Tbet expression and failed to produce IFNγ. Moreover, in contrast to previously described ILC1 subsets they could be efficiently differentiated into NK cells. These were characterized by highly diversified KIR repertoires including late stage NKG2A⁻KIR⁺ effector cells that are commonly not generated from previously known NK cell progenitor sources. This property was dependent on stroma cell-derived Notch ligands. The frequency of the novel ILC1-like NK cell progenitor (NKP) significantly declined in CB from early to late gestational age. The study supports a model in which circulating fetal ILC1-like NKPs travel to secondary lymphoid tissues to initiate the formation of diversified NK cell repertoires after birth.

**\*For correspondence:**
Markus.uhrberg@med.uni-duesseldorf.de

**Competing interests:** The authors declare that no competing interests exist.

## Introduction

Innate lymphoid cells (ILC) constitute a novel family of non-B, non-T cell lymphocytes that was established within the last decade (*Spits et al., 2013*; *Vivier et al., 2018*). The ILC nomenclature mirrors previously established T cell effector definitions and can be divided into three functional groups: natural killer (NK) cells and ILC1 are grouped within group 1 ILCs due to their expression of the transcription factor (TF) T-bet and secretion of IFNγ (*Bernink et al., 2013*; *Spits et al., 2013*); ILC2 belong into ILC group 2 and produce $T_H2$-like cytokines as well as express the TF GATA-3 (*Mjösberg et al., 2011*); ILC3 as well as fetal lymphoid tissue inducer cells belong into group 3 ILC, they secrete IL-17 and/or IL-22 and depend on TF RORγt expression (*Hoorweg et al., 2012*). In humans, non-NK ILCs are conventionally defined and physically enriched on the basis of IL-7 receptor expression (CD127) in combination with the exclusion of a lineage marker panel (*Bianca Bennstein et al., 2019*; *Krabbendam et al., 2018*; *Spits et al., 2013*; *Vivier et al., 2018*). Further differentiation into ILC subgroups involves the presence of CD117 on ILC3, of CRTH2 on ILC2, and the lack of both in case of ILC1.

The transcriptional and functional identity of ILC1 in humans is still a matter of debate, which is partly due to the fact that in contrast to other ILC subsets, ILC1 are lacking robust markers enabling their positive identification and isolation. Within the original description, ILC1 were defined as lin⁻CD127⁺CD117⁻CRTH2⁻CD161⁺ cells (*Bernink et al., 2013*). However, CD161 is also an NK cell 'marker' and CD127 is consistently expressed on the CD56^bright NK cell subset (*Vivier et al., 2018*).

Nonetheless, ILC1 can be robustly separated from NK cells in most settings by the lack of the CD94/NKG2A heterodimer and/or KIR, representing the prime receptors for missing self-recognition, thereby constituting an exclusive phenotypic and functional hallmark of NK cells (*Bernink et al., 2013*; *Manser et al., 2015*). In addition to the original described ILC1 subset, an intraepithelial CD103+Eomes+ type 1 ILC has been described expressing T-bet and secreting IFNγ (*Cella et al., 2019*; *Fuchs et al., 2013*). Of note, the intraepithelial CD103+ subset expresses typical NK cell markers such as CD94, NKG2A, and granzymes (*Cella et al., 2019*; *Krämer et al., 2017*). Since CD103 is an established tissue residency marker frequently found on NK cells within mucosa-associated lymphoid tissues (*Freud et al., 2017*) and moreover transcriptional as well as phenotypic analysis failed to clearly separate CD103+ intraepithelial ILC1 from NK cells (*Cella et al., 2019*; *Yudanin et al., 2019*), the CD103+ intraepithelial subset appears to be a tissue-resident NK cell subset.

Regarding the developmental relationship between human NK cells and ILC1, recent data support the existence of separate precursors for the development of ILC1 and NK cells downstream of the common lymphoid progenitor stage (*Renoux et al., 2015*; *Vivier et al., 2018*), which is a revision of the initial model assuming a direct common progenitor of ILC1 and NK cells (*Spits et al., 2013*). Nevertheless, the question remains what relationship the two human type 1 ILC types have to one another. Several studies in mice suggest a conversion of NK cells into ILC1 (*Cortez et al., 2017*; *Park et al., 2019*). In contrast, it is currently not known if ILC1 can be converted into NK cells, except by reprogramming of murine ILC1 with Eomes, a central TF for NK cell development and maturation (*Pikovskaya et al., 2016*). Notably, plasticity between NK and ILC1s has so far not been shown in humans.

So far, definitions of ILC1 are predominantly based on work in solid organs and tissues such as gut and lymph nodes (*Bernink et al., 2013*; *Björklund et al., 2016*). In contrast, for circulating ILC1, information on origin, function, and developmental potential is still at its infancy. Given that the accessibility of ILCs in humans is mostly restricted to blood, an increased understanding of the biology of blood-borne ILC and their respective progenitors appears to be of pivotal importance for guiding and implementing future ILC-based cellular therapies. Of note, in PB it was recently shown that lin-CD117+ cells, phenotypically resembling ILC3, were observed to be functionally immature but instead could be differentiated into all ILC subsets including NK cells (*Lim et al., 2017*).

In the present study, a thorough characterization of circulating type 1 ILC was performed in human umbilical cord blood (CB). CB represents a highly versatile and ethically non-problematic source of neonatal blood with low pathogenic burden that was recently shown to be enriched for ILCs compared to PB (*Vély et al., 2016*). Our work demonstrates that CB-derived ILC1-like cells are distinct from NK cells on the transcriptional, epigenetic, and functional level but rather constitute NK cell progenitors (NKP) with a unique propensity to generate clonally diversified NK cell repertoires in vitro. A similar ILC1-like subset, albeit at lower frequency, was also found in peripheral blood.

## Results

### Distinct transcriptional identities of neonatal circulating ILC1-like cells and NK cells

The transcriptional basis underlying the phenotypic and functional differences between NK cells and ILC1, together comprising the group 1 ILC family, is poorly defined in the circulation and direct comparisons between ILC1 and NK cells by bulk RNA sequencing are so far not available. A central purpose of the present study was to characterize group 1 ILC in CB, which provides a rich source for ILCs comprising all three ILC subsets as well as both major NK cell subsets, the regulatory CD56bright and the cytotoxic CD56dim NK cells (*Bianca Bennstein et al., 2019*). To this end, ILC1-like cells, defined as lin-CD94-CD127+CRTH2-CD117- cells, CD56bright NK cells, and CD56dim NK cells were flow cytometrically sorted from freshly collected CB and subjected to RNAseq analysis (*Figure 1— figure supplement 1*). All three group 1 ILC subsets could be clearly separated from each other on the basis of their transcriptional patterns in the heatmap and principal component analysis (*Figure 1a,b*). PC1, which accounts for 56% variance in the data, differentiates between the three cell subsets with ILC1-like cells clearly separated from CD56dim NK cells and CD56bright NK cells. Within PC2 accounting for 16% of variance, all three subsets could be further separated from each

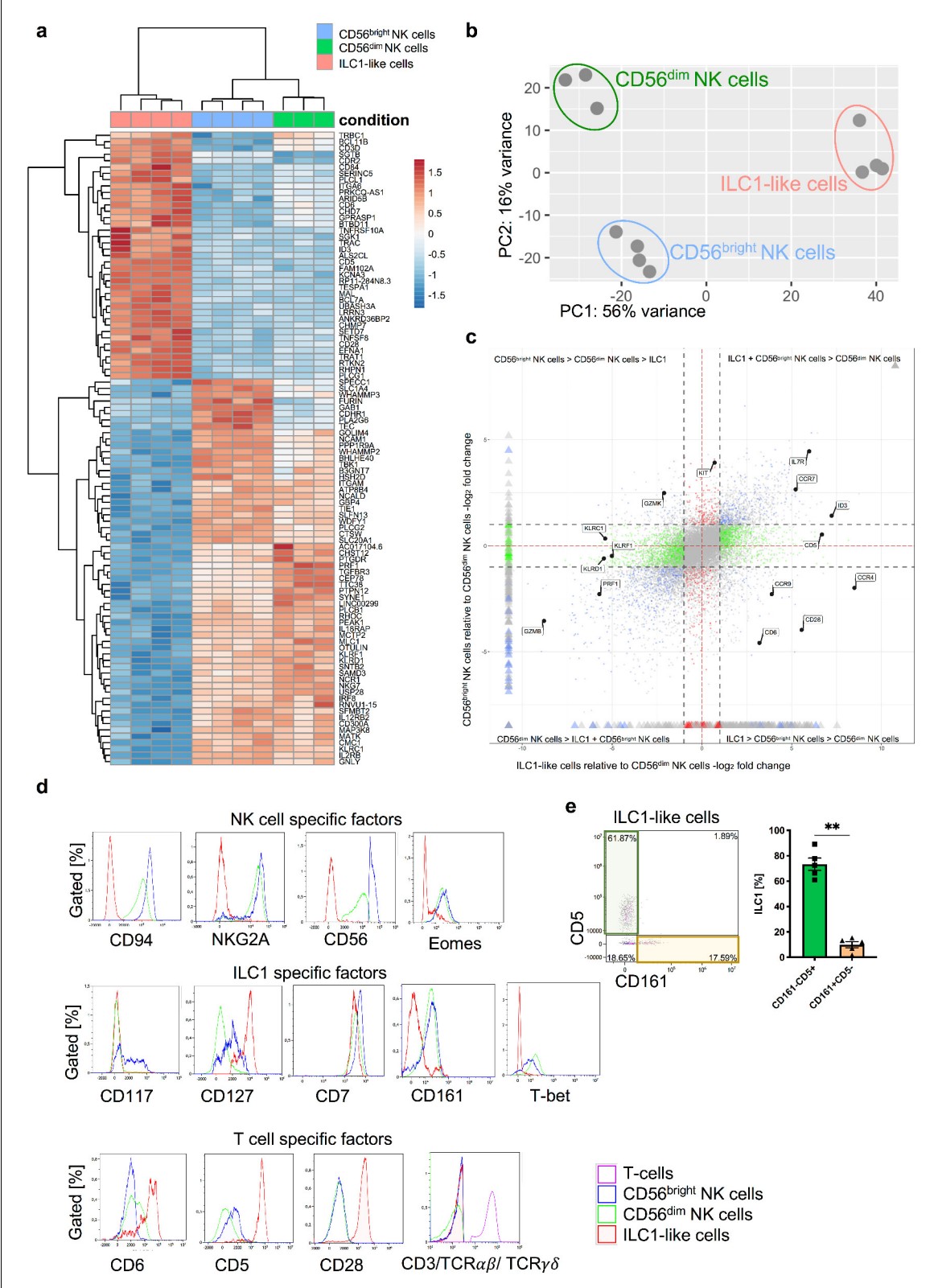

**Figure 1.** ILC1-like cells have a unique gene signature distinct from NK cells. (a) CB mononuclear cells (MNCs) were enriched prior to sorting via biotin-labelled antibodies (anti-CD3/CD19/CD14/CD66b) and sorted for ILC1-like cells, CD56$^{dim}$, and CD56$^{bright}$ NK cells (see *Figure 1—figure supplement 1* for sorting strategy). RNA sequencing was done on the Illumina platform. The heat map indicates the top 100 differentially expressed genes between ILC1-like cells and CD56$^{bright}$ NK cells including CD56$^{dim}$ NK cells. (b) A two-dimensional principle component analyses based on the top 2000

*Figure 1 continued on next page*

Figure 1 continued

differentially transcribed genes of CD56$^{bright}$ NK cells, CD56$^{dim}$ NK cells, and ILC1-like cells is shown. (c) A four-way plot with a cut off at a log$_2$ fold change ±1 (dotted lines) and adjusted p-values of 0.05 showing differently expressed genes of CB CD56$^{bright}$ NK cells compared to CD56$^{dim}$ NK cells and ILC1-like cells. Blue dots represent genes with an adjusted p-value<0.05 with a fold change >1. Green dots represent genes with an adjusted p-value <0.05 with a fold change between >1 (x-axis) and <1 (y-axis). Grey dots represent genes with an adjusted p-value >0.05. Red dots represent genes with an adjusted p-value<0.05 with fold rates < 1 (x-axis) and >1 (y-axis). Selected genes differentially expressed between NK cells subsets and ILC1-like cells are highlighted. (d) CB MNCs (n = 3) gated on ILC1-like cells, CD56$^{bright}$ NK cells, and CD56$^{dim}$ NK cells, respectively were analyzed by flow cytometry for selected NK, T, and ILC markers. Representative histograms for NK cell-specific factors containing CD94, NKG2A, CD56, and EOMES (upper panel), ILC1-specific factors containing CD117, CD127, CD7, CD161, and TBET (middle panel), and T cell-specific factors CD6, CD5, CD28, as well as a Mix of CD3/TCRαβ/TCRγδ (bottom panel). (e) Representative dot plot of the expression of CD5 and CD161 within CB ILC1-like cells with representative quantification of CD5$^+$ (green bar and box) and CD161$^+$ (yellow bar and box), (n = 5). The height of the bar represents the mean ± SEM. Levels of significance were calculated with a non-parametric t test (Mann-Whitney), ** p-value <0.01. Data represent at least three different donors and experiments.

The online version of this article includes the following source data and figure supplement(s) for figure 1:

**Source data 1.** R code for RNAseq data analyses used in *Figure 1*.
**Figure supplement 1.** Exemplary gating strategy for ILC and NK cell sorting.
**Figure supplement 2.** CD161$^+$ILC1-like cells and CD5$^+$ILC1-like cells differ in their CD28 and CD6 expression.
**Figure supplement 3.** ILC1-like cells phenotypically similar to T cells.
**Figure supplement 4.** Weak expression of EOMES and TBET in ILC1-like cells.

other. While in PC1 (56% variance) ILC1-like cells are more similar to CD56$^{bright}$ NK cells, in PC2 (16% variance) ILC1-like cells are more similar to CD56$^{dim}$ NK cells.

When analyzing the most differentially expressed genes (*Figure 1a,c–d*), ILC1-like cells were distinguished from NK cells by the lack of expression of typical NK cell markers such as CD56 (*NCAM1*), NKp46 (*NCR1*), NKp80 (*KLRF1*), NKG2A (*KLRC1*), and CD94 (*KLRD1*), low expression of receptor subunits for key innate cytokines IL-2 (*IL2RB*), IL-12 (*IL12RB2*), and IL-18 (*IL18RAP*), as well as lack of cytotoxic effector molecules perforin (*PRF1*), granulysin (*GNLY*), and all five members of the granzyme family. Whereas ILC1-like cells were apparently lacking basic NK cell characteristics, several of the most highly expressed genes within ILC1-like cells turned out to encode proteins associated with the T cell lineage including T cell surface markers CD5, CD6, and CD28 (*Figure 1a,c–d*). Furthermore, specific components of the T-cell receptor (TCR) unit such as TCRB constant chain (*TRBC1*) and CD3δ (*CD3D*) were more strongly transcribed in ILC1-like cells, albeit moderate transcription was also present in NK cells, particularly the more mature CD56$^{dim}$ subset (*Figure 1a*). Nevertheless, ILC1-like cells lacked surface expression of CD3, TCRαβ, and TCRγδ (*Figure 1d*). On the basis of CD161 that has been previously described to be expressed on tonsillar ILC1 (*Bernink et al., 2013*) and the T cell marker CD5, ILC1-like cells could be further subdivided into two main subsets, a major CD5$^+$CD161$^-$ subset co-expressing other T cell lineage markers such as CD6 and CD28 and a minor CD5$^-$CD161$^+$ population dominantly lacking CD6 with moderate CD28 expression (*Figure 1e* and *Figure 1—figure supplement 2*). We further characterized CB ILC1-like cells according to their T cell-associated characteristics and found expression of either CD4 or CD8, intracellular CD3δ, selective CD2 expression on CD5$^+$ ILC1-like cells, and a polyclonal repertoire of *TRAV* and *TRBV* encoding the variable regions of the T cell receptor alpha and beta chain, respectively (*Figure 1—figure supplement 3*).

We next analyzed the expression of lineage-determining transcription factors. Inhibitor of DNA binding 3 (*ID3*), a helix-loop-helix (HLH) protein that is generally expressed in the T cell but not NK cell lineage was found to be highly transcribed in ILC1-like cells but not NK cells, again pointing toward a closer relationship of ILC1-like cells to T cells (*Figure 1a,c*). Furthermore, Eomesodermin (*Eomes*) encoding a key transcription factor for NK cell development, was highly expressed in NK cells, but also found to be moderately expressed in ILC1-like cells by RNAseq analysis and also by intranuclear staining (*Figure 1d*, *Figure 1—figure supplement 4*). Finally, TBET (*TBX21*), originally reported to be a defining feature of both, ILC1 and NK cells, was strongly expressed on EOMES$^+$ NK cells, whereas it was almost absent on ILC1-like cells (*Figure 1d*, *Figure 1—figure supplement 4*). Together, we show that neonatal circulating ILC1-like cells have a unique transcriptional identity distinct from NK cells and on the other hand exhibiting phenotypic similarities with T cells including expression of lineage markers, TCR components, and transcription factors.

## Expression of chemotactic receptors suggests differential migratory behavior of ILC1-like subsets and NK cells

Among the most significant changes identified by transcriptional analysis between ILC1-like and NK cells were the chemokine receptors *CCR7*, which plays a key role in promoting migration to secondary lymphoid organs, as well as *CCR4* and *CCR9* which are involved in migration to skin and small intestine, respectively (*Oo and Adams, 2010*; *Figures 1c* and *2a*). All three receptors were prominently expressed in ILC1-like cells but lacking or weakly expressed in CD56$^{dim}$ and CD56$^{bright}$ NK cells, respectively. The chemokine receptor pattern of the ILC1-like subset was reminiscent of peripheral T cells and suggests fundamentally different migratory properties of ILC1-like cells under steady-state conditions compared to circulating NK cells. In contrast, NK cells but not ILC1-like cells exhibited high levels of the Sphingosine-1 phosphate (S1P) receptor *S1PR5*, which is a potent chemotactic regulator of tissue residency (*Figure 2a*) suggesting that circulating neonatal NK cells, in particular CD56$^{dim}$ cells, are more bound to stay within the circulation compared to ILC1-like cells.

In accordance with the RNAseq data (*Figure 1c*), CCR7 was highly expressed on the cell surface of ILC1-like cells but barely detectable on NK cells (*Figure 2b*). In case of CCR9, high surface expression was found on small subsets of CCR7$^+$ and CCR7$^-$ ILC1-like cells (12.97% vs 8.74%) but not NK cells. We further observed a distinct CCR4$^+$ILC1-like subset (15.72%), but no CCR4 expression on NK cells (*Figure 2b*). Further analysis of the distribution of chemokine receptors on the subsets defined by CD5 and CD161 expression shown above (*Figure 1e*) by t-distributed stochastic neighbor embedding (t-SNE) analysis revealed that CCR4, CCR7 and CCR9 expression was restricted to CD5$^+$ ILC1-like cells, (either CD5$^+$CD161$^-$ or CD5$^+$CD161$^+$), whereas the small CD5$^-$CD161$^+$ subset did not express any of the three chemokine receptors (*Figure 2c–d*). The data thus demonstrate that the majority of ILC1-like cells (CD5$^+$CD161$^{+/-}$) express chemokine receptors enabling migration into various tissues, whereas a small subset (CD5$^-$CD161$^+$), similar to NK cells, lack this property.

## IFNγ production is restricted to CD161$^+$ ILC1-like cells

A key effector function of ILC1 cells is the rapid production of IFNγ in response to inflammatory cytokines. Unexpectedly, only very few ILC1-like cells exhibited intracellular IFNγ production (mean: 1,25%), whereas the large majority of CD56$^{bright}$ NK cells readily produced IFNγ as expected (mean: 80,3%), (*Figure 2e*). Similarly, upon polyclonal stimulation with PMA/Ionomycin, ILC1-like cells were again largely unable to produce IFNγ (*Figure 2f*). Even over an extended period of five days, ILC1-like cells produced very low amounts of IFNγ, again in contrast to NK cells (*Figure 2g*). Importantly, differential effector functions were noted when comparing the major CD5$^+$CD161$^-$ and the minor CD5$^-$CD161$^+$ subsets. The CD161$^+$ subset contained a small fraction of cells able to produce IFNγ after short-term cytokine stimulation, whereas IFNγ-producing cells were almost undetectable in the major CD161$^-$ subset (mean: 8.2% vs. 0.9%), (*Figure 2h*). Together, the data suggest that the majority of ILC1-like cells (CD5$^+$CD161$^-$) are functionally immature but that a minor subset of CD5$^-$CD161$^+$ cells exerts IFNγ-mediated effector functions.

## Decline of circulating ILC1-like cells with gestational age

The pattern of chemokine and S1P receptor expression suggested fundamentally different migratory properties of neonatal ILC1-like cells compared to circulating NK cells under steady-state conditions. In particular, CCR7 expression was most prominent in ILC1-like cells suggesting their efficient migration to secondary lymph nodes. In order to better understand the dynamics of ILC subsets in the circulation around birth, ILC frequencies were analyzed in CB according to gestational age. As shown in *Figure 3*, CD56$^{bright}$ and CD56$^{dim}$ NK cell frequency as well as total cell count seemed largely unaffected by changes in gestational age. In contrast, a significant decrease of ILC1-like cells in terms of frequency (r = 0.9286, p-value = ** 0.0067) and total cell number (r = 0.8929, p-value = *0.0123) was found with increasing gestational age of the CB. The decline was specific for ILC1-like cells and not observed for ILC2 and ILC3 subsets, which showed no significant changes in either direction by gestational age. Analysis of ILC frequencies in adults revealed no further decline of ILC1-like cells in young and middle-aged (18–55 years) or elderly (63–86 years) adults. In contrast, a strong decline was observed for ILC2 and ILC3 subsets from neonatal to adults with an additional significant decrease from middle-aged to elderly adults (*Figure 3*). Thus, transcriptomic, phenotypic,

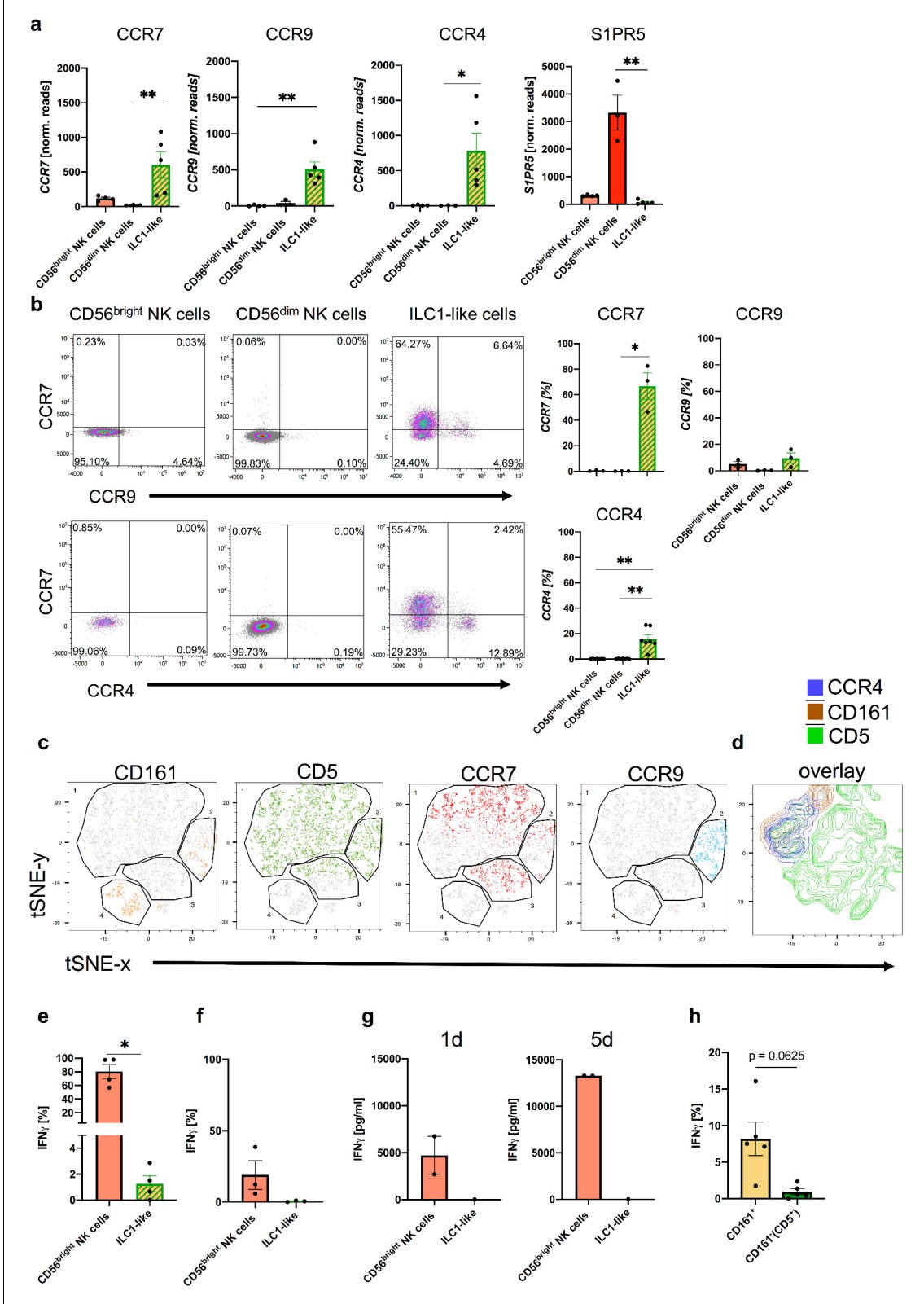

**Figure 2.** CD5[+] and CD161[+] ILC1-like subsets are distinguished by differential chemokine receptor expression and functionality. (**a**) Expression of CCR7 (left corner), CCR9 (left middle), CCR4 (right middle), and S1PR5 (right corner) determined by RNA sequencing for CD56[bright] NK cells (n = 4), CD56[dim] NK cells (n = 3), and ILC1-like cells (n = 5). (**b**) Surface expression of chemokine receptors on CD56[bright] NK cells, CD56[dim] NK cells, and ILC1-like cells in ex vivo isolated MNC from CB. Representative dot plots and quantification of CCR7 and CCR9 (n = 3) or CCR7 and CCR4 (n = 7) is shown. (**c and d**) *Figure 2 continued on next page*

*Figure 2 continued*

t-SNE plots for expression of CD161, CD5, CCR7, and CCR9 as well as an overlay of CD161, CD5, and CCR4 expression (rightmost panel) on ILC1-like cells (n = 3) calculated with 500 iterations (see *Figure 1—figure supplement 1* for gating of ILC1-like cells). (e and f) Freshly isolated CB MNC were either stimulated with IL-12 (5 ng/ml) and IL-18 (50 ng/ml) overnight or with PMA/Ionomycin for 4 hr to measure intracellular expression (n = 5/3). (g) CB ILC1-like cells were sorted and stimulated with IL-12/IL-18. At day 1 and 5 supernatant was taken and analysed for IFNγ secretion (n = 1–2). (h) MNCs stimulated with IL-12/IL-18 were further gated on CD161⁻ and CD161⁺ cells and IFNγ secretion was determined. The heights of the bars represent the mean ± SEM. Levels of significance were calculated with a One-Way ANOVA with a multiple correction post-test (Kruskal-Wallis test) (a and b), by a Mann-Whitney test (e–g) and Wilcoxon ranked test (h), * p-value<0.05, ** p-value<0.01, *** p-value<0.001. Data represent at least three different donors (a–f, h) as well as one to two donors (g) and two experiments.

and age-related analysis suggest a unique as well as highly dynamic role of ILC1-like cells in the circulation before and around the time of birth.

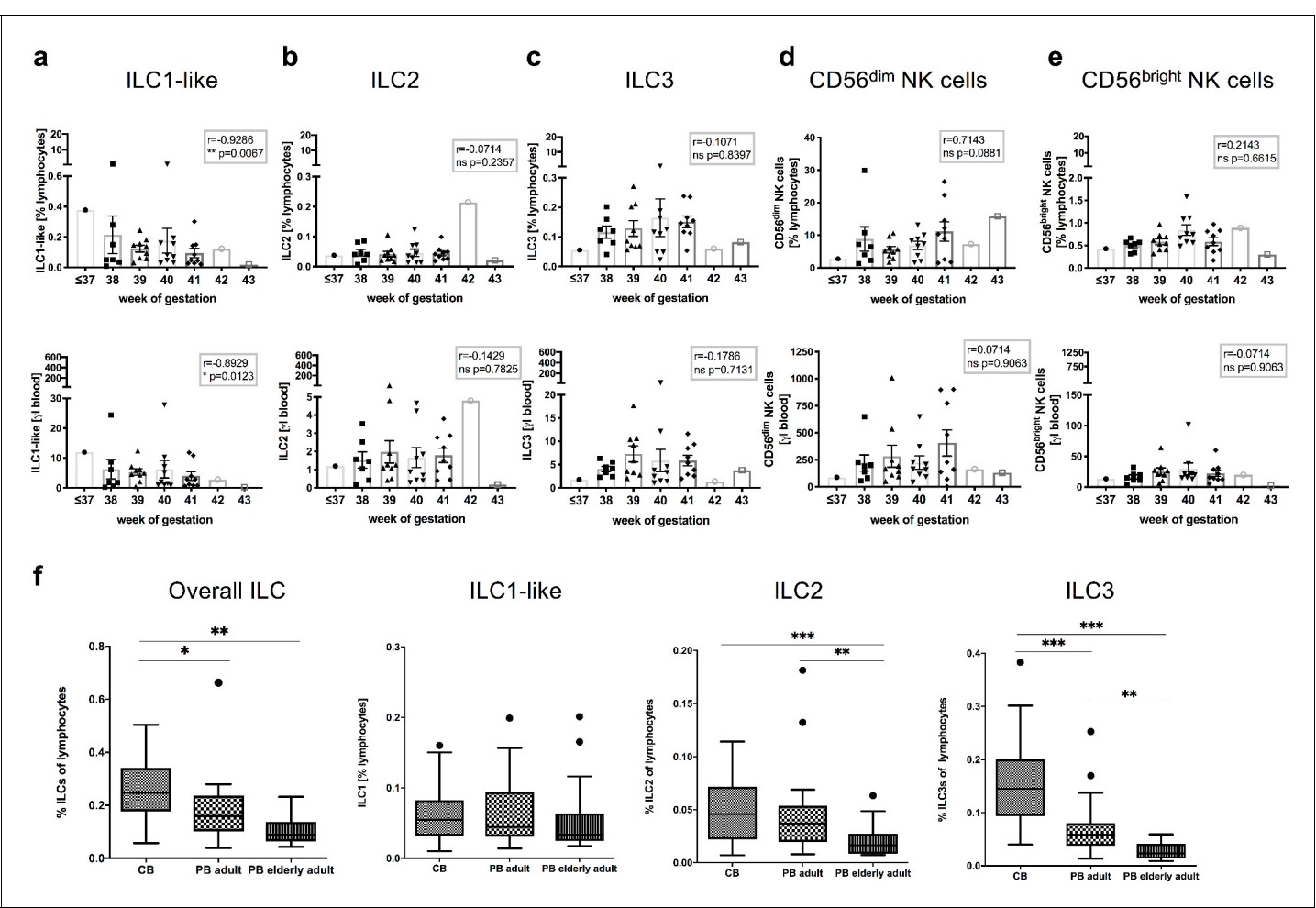

**Figure 3.** Frequency and cell count of ILC1-like cells from CB inversely correlates with gestational age. (a–e) Frequency (top panel) and cell count per µl blood (lower panel) of CB samples (n = 37) according to gestational age are shown from left to right for ILC1-like cells, ILC2, ILC3, CD56ᵈⁱᵐ NK cells, and CD56ᵇʳⁱᵍʰᵗ NK cells. (f) Frequency of ILCs within lymphocytes from CB (n = 32), adult PB (n = 22, age 18–55 years), and elderly PB (n = 20, age 63–86) as Tukey box plots from left to right for total ILC (Lin⁻CD94⁻CD45⁺CD127⁺), ILC1-like cells (CD117⁻CRTH2⁻), ILC2 (CD117⁻/⁺CRTH2⁺), and ILC3 (CD117⁺CRTH2⁻). The heights of the bars represent the mean ± SEM. Levels of significance were calculated using a Spearman correlation (a–e) and a Kruskal-Wallis test with a Mann-Whitney U post-test and Bonferroni corrected p-values for multiple testing (f), * p-value<0.05, ** p-value<0.01, *** p-value<0.001. Data represent at least three different donors and experiments.

## Neonatal ILC1-like cells contain a novel NK cell progenitor

Based on the observation that the large majority of neonatal ILC1-like cells are functionally immature, we next explored the possibility that they constitute a novel type of lymphoid progenitor. To this end, the two main ILC1-like subsets expressing CD5$^+$ (CD161$^{-/+}$) or lacking CD5$^-$ (CD161$^{-/+}$) were seeded onto the stromal feeder cell line OP9-DL1 that is well described to support differentiation into the NK cell as well as the T cell lineage depending on the respective cytokine conditions (*Freud et al., 2006*; *Schmitt and Zúñiga-Pflücker, 2002*; *Figure 4a*). The ILC1-like subsets were compared to CD56$^{bright}$ NK cells constituting a well-described immediate progenitor of mature NK cells and to ILC2 cells (lin$^-$CD94$^-$CD127$^+$CRTH2$^+$), representing an innate lymphocyte subset supposedly lacking NK cell differentiation potential. When ILC1-like cells were subjected to T cell differentiation conditions (*Wang et al., 2006*), very few cells survived the first 8 days of differentiation (*Figure 4—figure supplement 1*) and no CD3$^+$ T cells could be detected suggesting that under these conditions ILC1-like cells do not efficiently differentiate into the T cell direction. In contrast, when subjected to NK cell differentiation conditions, CD5$^+$ as well as CD5$^-$ ILC1-like cells upregulated CD94 and NKG2A expression de novo. Remarkably, ILC1-like cells more efficiently differentiated into mature KIR$^+$ NK cells than CD56$^{bright}$ NK cells, which largely maintained their initial NKG2A expression (*Figure 4b*). ILC2 remained negative for CD94 and NKG2A receptors. To further characterized the in vitro generated NK cells, we stained for CD56 and NKp46 expression, which are two additional phenotypic 'markers' of NK cells as well as stained intracellularly for perforin, granzyme B, and granzyme K expression. Not surprisingly, we observed a significant higher expression of CD56 on cultured CD56$^{bright}$ NK cells compared to ILC1-like cells (94.50% vs 78.23%, ** p-value: 0.0022). We further observed slightly higher NKp46 expression in cultured ILC1-like cells compared to CD56$^{bright}$ NK cells and a similar expression pattern of perforin, granzyme B, and granzyme K of both cultured cell populations to ex vivo CD56$^{dim}$ NK cells (*Figure 4—figure supplement 2*) suggesting that both cell types acquired cytotoxic potential.

Based on the rapid upregulation of NK cell receptors on ILC1-like cells in culture we were wondering if the respective genes were already epigenetically poised for transcription. To this end, we analyzed sorted neonatal ILC1-like cells as well as NK cells by ATACseq, constituting a sensitive global method to assess chromatin accessibility, which serves as correlate for epigenetic remodeling of the locus. Whereas in NK cells, the CD94, NKG2A, and KIR2DL3 genes exhibited highly accessible chromatin regions around the transcriptional start points, as expected, in ILC1-like cells only moderate (CD94) or no signs of chromatin remodeling (NKG2A, KIR2DL3) were found (*Figure 4c*), excluding epigenetic conditioning toward expression of these NK cell receptors. In contrast, the T cell-specific marker CD5 exhibited open chromatin structures in the 5'-regulatory regions in ILC1-like cells whereas they were inaccessible in NK cells. In line with the expression of CD161 on NK cells and only in a small fraction of ILC1-like cells, we observed more open chromatin structures in the 5'-region in NK cells compared to ILC1-like cells (*Figure 1—figure supplement 2c*).

## NK cells derived from neonatal ILC1-like cells are functionally mature

We next assessed the functional properties of the putative NK cell population generated from neonatal ILC1-like cells in vitro. Analysis of CD107a mobilization, representing a correlate for the degranulation of cytotoxic granules, revealed a high frequency (mean: 61.1%) of CD107a$^+$ ILC1-like-derived NK cells upon incubation with the HLA class I-deficient target cell line K562, comparable to the results with CD56$^{bright}$ NK cells (*Figure 5a*). Assessment of direct cytotoxicity similarly revealed comparable effector functions of NK cells derived from ILC1-like cells or CD56$^{bright}$ NK cells (*Figure 5b*). Furthermore, NK cells derived from ILC1-like cells showed upregulation of CD16, constituting an important Fc receptor type for mediating antibody-dependent cellular cytotoxicity (ADCC) (*Figure 5c*). When incubating the in vitro-generated NK cells with the CD20-specific antibody Rituximab and the CD20$^+$ B cell line Raji, a comparable specific ADCC function was observed in NK cells derived from ILC1-like cells and CD56$^{bright}$ NK cells (mean: 6.7% ILC1-like-derived NK cells vs. 13.6% CD56$^{bright}$-derived NK cells) (*Figure 5d*). Together, NK cells from ILC1-like cells exhibit key NK cell effector functions including mobilization of cytotoxic granules, killing of HLA-deficient target cells, and CD16-mediated ADCC.

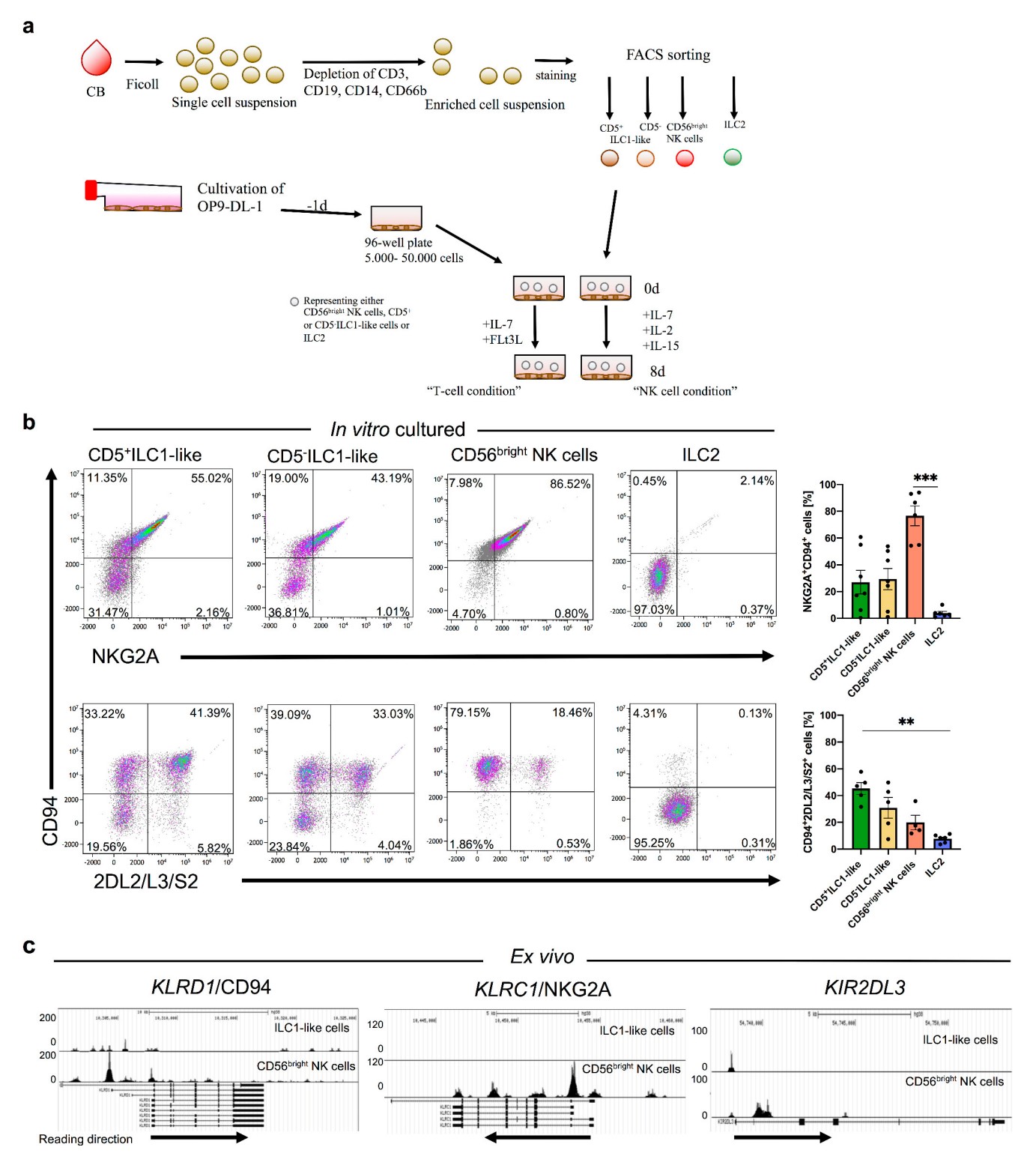

**Figure 4.** ILC1-like cells possess high NKP potential without previous epigenetic priming for NK cell receptor expression. (**a**) Scheme of the experimental set-up. CB MNCS were freshly isolated, enriched using biotinylated antibodies (anti-CD3, CD14, CD19, CD66b), and sorted for CD5+ILC1-like cells, CD5-ILC1-like cells, CD56bright NK cells, and ILC2. One day prior to sorting, OP9-DL1 cells were plated in 96-well flat-bottom plates. Cells were either supplemented with IL-7 and FLt3L for the T cell condition or IL-2, IL-7, and IL-15 for the NK cell condition and cultured for 8 days with medium change at day 5. (**b**) Exemplary dot plots and quantification of CD94 expression together with either NKG2A or KIR2DL2/L3/S2 expression after

*Figure 4 continued on next page*

Figure 4 continued

8 days of co-culture on OP9-DL1 from left to right for CD5⁺ILC1-like cells (n = 7), CD5⁻ILC1-like cells (n = 7), CD56$^{bright}$ NK cells (n = 5), and ILC2 (n = 6). (c) Comparative analysis of regions with open chromatin by ATAC sequencing for *KLRD1* (CD94), *KLRC1* (NKG2A), and *KIR2DL3*. For ATAC sequencing, 5000 CB-derived ILC1-like (top row) and NK cells (bottom row) were flow cytometrically sorted to >99% purity (n = 3). Arrows underneath the ATAC data indicate orientation and start of gene transcription. Heights of the bars represent mean ± SEM. Levels of significance were calculated with a One-Way ANOVA with a multiple comparison post-test (Kruskal-Wallis test), * p-value<0.05, ** p-value<0.01, *** p-value<0.001. Data represent at least three different donors and experiments (a–b). Data represent two experiments with three donors (c).

The online version of this article includes the following figure supplement(s) for figure 4:

Figure supplement 1. ILC-1-like cells proliferate in NK cell but not T cell conditions.
Figure supplement 2. ILC1-like-derived NK cells express essential NK cell characteristics: CD56, NKp46, Perforin, and Granzyme B.

## ILC1-like cells acquire KIR receptors in a NOTCH-dependent manner

Others and we had previously shown that the presence of Notch ligands such as delta ligand 1 (DLL1) in the hematopoietic niche plays a key role in instructing NK cell progenitors for later KIR expression (*Miller et al., 1999*; *Zhao et al., 2018*). In order to evaluate a possible role for NOTCH ligands in our system, experiments were repeated in a purely cytokine-based environment as well as on OP9 stromal cells lacking DLL1 expression and then compared to the original conditions using the DLL1-transfected OP9-DL1. The experiments using sorted CB-derived ILC1-like cells and CD56$^{bright}$ NK cells revealed that stroma cells are generally promoting NK cell differentiation of ILC1-like cells (*Figure 6*), as previously seen for established stage 2 (CD34⁺CD117⁺) NK cell progenitors. CD94⁺NKG2A⁺ NK cells were generated with highest frequency on OP9, followed by OP9-DL1 cultures and with only low frequency in stroma free conditions (*Figure 6a–b*). Thus, NOTCH signaling seemed to have no promoting influence on the generation of CD94⁺NKG2A⁺ NK cells. Furthermore, CD56$^{bright}$ NK cells largely maintained CD94⁺NKG2A⁺ expression in all three conditions, as expected. In contrast, the presence of DLL1-transfected stroma cells had a significant influence on KIR expression: OP9-DL1 cells efficiently supported the generation of KIR2DL2/3⁺ NK cells whereas on OP9 stroma cells lacking DLL1 only few KIR2DL2/3⁺ NK cells were generated. This effect was similarly seen for CD56$^{bright}$ NK cells, albeit on a lower quantitative level (*Figure 6a–b*).

## ILC1-like progenitors from CB and PB support the generation of highly diversified NK cell repertoires

In order to more thoroughly assess the NK cell differentiation potential of ILC1-like cells, we surveyed the complexity of the in vitro generated NK cell repertoires by analyzing expression of NKG2A together with KIR2DL1, KIR2DL2/2DL3, and KIR3DL1, representing the major inhibitory receptors for the four HLA class I-encoded epitopes E, C2, C1, and Bw4, respectively. Furthermore, KIR genotypes were determined to evaluate the presence/absence polymorphism of KIR genes in each individual (*Figure 6—figure supplement 1*). A prominent population of NKG2A⁻KIR⁺ cells, representing a particularly late stage of NK cell development, was detected in cultures from ILC1-like cells but were almost absent in CD56$^{bright}$-derived NK cells (mean: 19.8% vs. 1.6%, p-value: 0.002) (*Figure 6c–d*). Moreover, a higher frequency of NK cells expressing any of the four inhibitory KIR was observed from ILC1-like cells compared to CD56$^{bright}$-derived NK cells (mean: 42.9% vs. 12.7%, p-value: 0.004) (*Figure 6d*). A similar picture emerged already after 8 days of co-cultivation (*Figure 6d/e*), suggesting a rapid differentiation process. Comparing the NKG2A/KIR repertoires of days 8 and 14, we observed slightly higher NKG2A⁺KIR⁻ and decreased NKG2A⁻KIR⁻frequencies in ILC1-like derived NK cells (NKG2A⁺KIR⁻: 23.6% vs. 33.7% (p-value: 0.121); NKG2A⁻KIR⁻: 54.8% vs. 39.2% (p-value: 0.0649). In line with a progression in differentiation, we detected in ILC1-like derived NK cells an increase of NKG2A⁺KIR2DL2/3⁺ or triple positive NKG2A⁺KIR2DL2/3⁺KIR3DL1⁺ from days 8 to 14. Furthermore, we detected no difference of NKG2A⁻KIR2DL2/L3⁺ cells between day 8 and 14, but slightly higher frequencies of mature NK cells expressing two KIRs (KIR3DL1⁺KIR2DL1⁺ and KIR2DL2/L3⁺KIR3DL1⁺), again indicating further differentiation (*Figure 6e*). ILC1-like cells created a much more diversified NK cell repertoire compared to CD56$^{bright}$ NK cells. All possible clonal receptor combinations were generated and all KIR⁺ clonotypes lacking NKG2A and thus representing advanced steps of NK cell differentiation were more frequently found in cultures from ILC1-like cells than from CD56$^{bright}$ NK cells (*Figure 6e*).

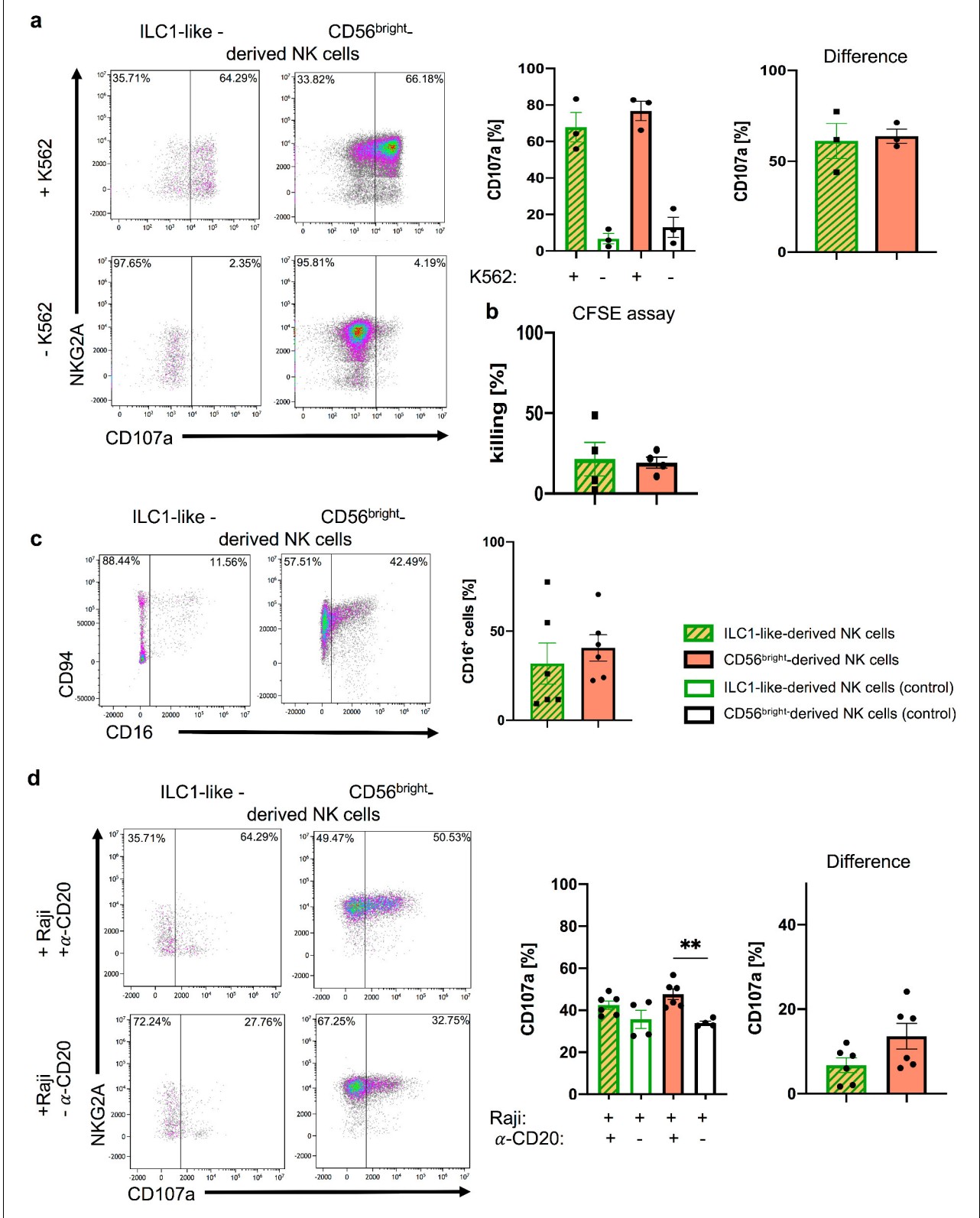

**Figure 5.** ILC1-like NKPs develop into effector NK cells. After 15 days of co-cultivation on OP9-DL1 stromal cells, NK cells derived from ILC1-like cells or CD56[bright] NK cells from CB were used in (**a**) granule mobilization (n = 3) and (**b**) cytotoxicity assays (n = 4) against the HLA-deficient cell line K562 at an effector/target ratio of 1:1. For CD107a quantification (a/d), analysis gates were set on cells expressing NKG2A and/or KIR. (**a**) Representative dot plots are shown for CD107a vs. NKG2A expression of NK cells derived from ILC1-like (left) or CD56[bright] NK cells (right) with K562 (top) and without

*Figure 5 continued on next page*

Figure 5 continued

K562 (bottom). Corresponding CD107a frequencies are shown as bar graphs with controls (left) and target-specific degranulation (right, calculated by subtracting spontaneous degranulation from degranulation with target cells) (n = 3). (c) Representative dot plots for CD94 and CD16 expression are shown for NK cells derived from ILC1-like cells (left) or CD56$^{bright}$ NK cells (right). Corresponding CD16 frequencies are shown as bar graphs (n = 6). (d) An ADCC assay was performed with CD20$^+$ Raji cells and Rituximab (anti-CD20) in an effector target ratio of 1:1. Exemplary dot plots for NKG2A vs. CD107a expression is shown for ILC1-like-derived NK cells (left) and CD56$^{bright}$-derived NK cells (right) cultured for 5 hr with Raji and Rituximab (top) or without Rituximab (bottom). Quantification of CD107a expression is shown for ILC1-like-derived NK cells and CD56$^{bright}$-derived NK cells with controls (left) and target-specific degranulation (right, calculated by subtracting degranulation without Rituximab from degranulation with Rituximab) (n = 4–6). Height of bars represent mean ± SEM. Levels of significance were calculated with a non-parametric two-tailed t test (Mann Whitney) and a One-Way ANOVA. Data points represent at least three different donors from at least two independent experiments (a–c) and one experiment with two different donors (d).

Next, we were wondering, if the observed developmental potential toward the NK cell lineage is similarly present in adult ILC1-like cells from peripheral blood. When cultivating flow cytometrically sorted PB ILC1-like cells on OP9-DL1 stroma cells, either no or only moderate NK cell expansion was detected (*Figure 6—figure supplement 2a–b*). Nonetheless, ILC1-like cells from PB, like their counterparts from CB, supported the development of mature NK cells, expressing significantly more KIR2DL2/2DL3 (mean: 49.9%) compared to PB CD56$^{bright}$-dervied-NK cells (mean: 13.5%) (*Figure 6—figure supplement 2c*). Furthermore, as already observed with CB, ILC1-like cells from PB were able to mature into NKG2A$^-$KIR$^+$ NK cells with higher frequency (mean: 14.7%) than CD56$^{bright}$-derived NK cells (mean: 3.6%) (*Figure 6—figure supplement 2d*).

## High clonogenic potential of ILC1-like cells toward generation of mature effector NK cells

We next analyzed the differentiation potential of neonatal ILC1-like cells on the clonal level. To this end, single cells from the three major ILC1-like populations defined above (CD161$^+$CD5$^-$, CD161$^-$CD5$^+$, CD161$^+$CD5$^+$) as well as from CD56$^{bright}$ NK cells were flow cytometrically deposited and cloned on OP9-DL1 cells. The cloning efficiency at day 14 was highest for CD56$^{bright}$ NK cells (mean: 50%), followed by CD161$^+$CD5$^+$ cells (mean: 35.2%), CD161$^+$CD5$^-$ (mean: 22.6%), and CD161$^-$CD5$^+$ (mean: 15.9%) (*Figure 7a*). The NKG2A$^-$KIR$^-$ subset, lacking *bona fide* NK cell markers was infrequent in all clonal cultures, ranging from 1–8% per clone thereby excluding efficient generation of any non-NK cells. Remarkably, the dominant population generated from CD161$^-$CD5$^+$ (mean: 92.3%) and to a lesser extent also from CD161$^+$CD5$^-$ ILC1-like cells (mean: 48.2%) were NKG2A$^-$KIR$^+$ NK cells, representing an advanced step of NK cell differentiation as outlined above. The population was less frequent in CD161$^+$CD5$^+$ clonal cultures (mean: 25.8%) and rare when starting from CD56$^{bright}$ cells (4.9%) (*Figure 7b*) consistent with the results from bulk differentiation experiments. CD56$^{bright}$ NK cells frequently lost their NKG2A and CD94 expression during clonal expansion (mean: 51.6%).

Stimulation of NK cell clones by K562 target cells led to increased granule mobilization as documented by the increased surface expression of CD107 (*Figure 7c*). Among ILC1-like subsets, NK cells from CD161$^+$CD5$^+$ cell showed the highest frequency of CD107a expression (mean: 49.2%), whereas the CD107a levels were comparable for clones generated from CD161$^+$CD5$^-$, CD161$^-$CD5$^+$, or CD56$^{bright}$ NK cell. Notably, elevated levels of CD107a were already observed in the absence of K562 for clones derived from ILC1-like cells (*Figure 7c*), which might be due to pre-activation by murine OP9-DL1 feeder cells constituting targets for the de novo generated NK cells (*Figure 7d*).

## Discussion

Here, we present the identification of an ILC1-like NKP within human CB and PB that is able to differentiate into mature cytotoxic NK cells. ILC1-like NKPs have a lin$^-$CD127$^+$CD117$^-$CRTH2$^-$ phenotype, matching innate lymphoid cells previously defined as ILC1 (*Bernink et al., 2013*), but lacking expression of TBET, a common signature transcription factor of ILC1 and NK cells. We demonstrate that ILC1-like cells can be broken down into two subsets, a major CD5$^+$CD161$^{-/+}$ subset expressing the chemokine receptors CCR7, CCR4, and CCR9 and failing to secrete IFNγ and a small CD5$^-$CD161$^+$ subset not expressing any of the three chemokine receptors but showing IFNγ

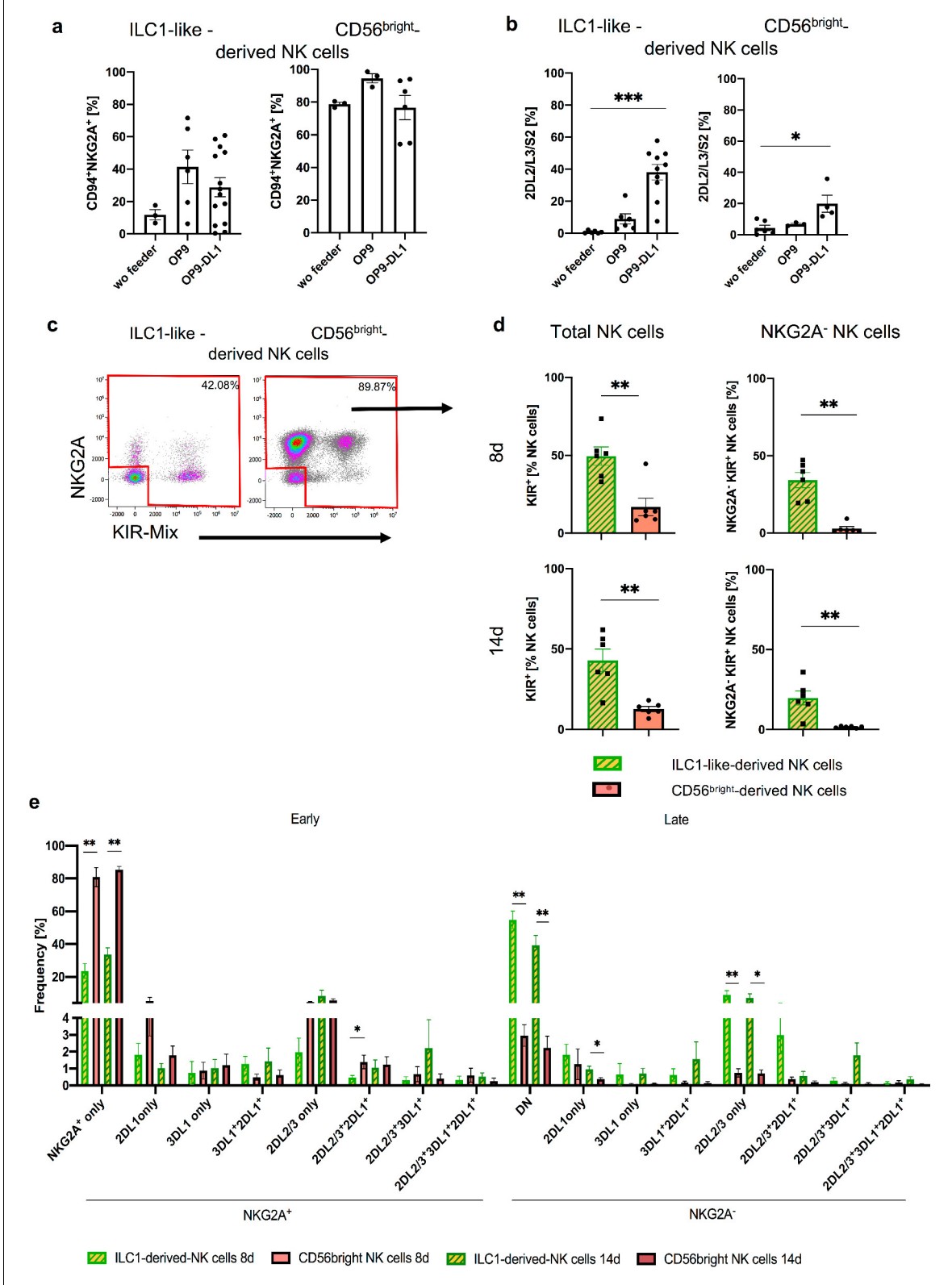

**Figure 6.** Differentiation of ILC1-like NKP lead to formation of complex NK cell repertoires via NOTCH signaling. CB-derived ILC1-like cells and CD56^bright NK cells were flow cytometrically sorted and subsequently cultured for 8 days on OP9, OP9-DL1, or without feeder cells (a, b) or cultured on OP9-DL1 for 8 and 14 days (c–e). (a) Frequency of CD94+NKG2A+ (n = 3–13) and (b) KIR2DL2/L3/S2+ NK cells (n = 3–9). (c) Representative dot plots for NKG2A and KIR (comprising antibodies against KIR2DL2/L3/S2, KIR2DL1/S1/S3/S5, and KIR3DL1) of ILC1-like-derived NK cells (left hand side) and

*Figure 6 continued on next page*

Figure 6 continued

CD56$^{bright}$-derived NK cells (right hand side). (d) Frequency of total KIR$^+$ (left hand side) and KIR$^+$NKG2A$^-$ (right hand side) NK cells derived from ILC1-like cells and CD56$^{bright}$ NK cells, respectively (n = 6) on day 8 (top) and day 14 (bottom). (e) Dissection of NK cell repertoire diversity of NK cells derived from ILC1-like cells and CD56$^{bright}$ NK cells, respectively by combinatorial analysis of the major inhibitory receptors KIR2DL1, KIR2DL2, KIR2DL3, KIR3DL1, and NKG2A at day 8 and 14 (n = 5–6, one donor missing KIR3DL1). DN refers to NKG2A$^-$KIR$^-$ NK cells. Color coding: ILC1-like-derived NK cells (day 8: light green and yellow, day 14: dark green with yellow) and CD56$^{bright}$-derived NK cells (day 8, light red, day 14 dark red). Height of the bars represent mean ± SEM. Levels of significance were calculated with a One-Way ANOVA with a multiple comparison post-test (Kruskal-Wallis test) (a, b) and an unpaired t test (Mann Whitney U) comparing both populations at the same time point. Data represent at least three independent experiments with each dot representing an individual donor (see *Figure 6—figure supplement 1* for individual KIR/NKG2A expression), * p-value<0.05, ** p-value<0.01, *** p-value<0.001.

The online version of this article includes the following figure supplement(s) for figure 6:

**Figure supplement 1.** Individual NK cell receptor repertoires after 8 and 14 days of co-culture on OP9-DL1.

**Figure supplement 2.** ILC1-like cells from PB have NKP potential.

secretion after specific stimulation. Despite their difference in phenotype and functionality, both ILC1-like subsets possessed NKP potential. Single-cell cloning experiments revealed a high NKP frequency within CB-derived ILC1-like cells and a high propensity to differentiate into mature NK cells. In contrast to established CD34$^+$ or CD34$^-$CD117$^+$ NKPs that predominantly generate NKG2A$^+$KIR$^-$ NK cells, differentiation of ILC1-like NKPs led to a high frequency of KIR$^+$ NK cells including the NKG2A$^-$KIR$^+$ subset, constituting an advanced maturation stage. The fact that neonatal ILC1-like cells are clearly distinct from NK cells on the transcriptomic, epigenetic, and functional level in combination with the single cell cloning experiments suggest that ILC1-like NKPs have a true progenitor relationship to NK cells.

Our study provides to our knowledge the first comparison of ILC1-like and NK cells by deep transcriptomic and epigenetic analysis in human blood. In the present study, ILC1-like cells could be clearly distinguished from CD56$^{bright}$ NK cells and the more mature CD56$^{dim}$ NK cells by RNAseq. The CB-derived ILC1-like cells completely lacked NK cell-specific molecules such as CD94, NKG2A, and KIR on the transcriptional and surface expression level. Moreover, chromatin accessibility studies suggested that the regulatory regions of the respective NK cell receptors were open in NK cells but closed in ILC1-like cells further confirming a lack of transcriptional activity for NK cell receptors in ILC1-like cells. The clear distinction between ILC1-like cells and NK cells contrasts with previous studies showing nearly indistinguishable expression patterns of the two group 1 ILC members (*Fuchs et al., 2013*; *Salomé et al., 2019*; *Yudanin et al., 2019*). In the case of CD127$^-$ILC, this is clearly a question of defining ILC1, since CD127$^-$ILC1 cells are different from circulating NK cells due to expression of CD103 but indistinguishable from tissue-resident NK cells, including the expression of EOMES and CD94/NKG2A NK cell receptors (*Fuchs et al., 2013*) and might thus as well be defined as NK cells. In case of CD127$^+$ILC1, contamination with CD56$^{bright}$ NK cells represents the main obstacle that has to be avoided by including CD94 in the lineage depletion cocktail (*Bernink et al., 2013*). Since the majority of CD56$^{bright}$ NK cells are CD127$^+$ and CD16$^-$, ILC1 subsets defined as CD56$^+$CD127$^+$ or CD56$^+$CD16$^-$ without exclusion of CD94$^+$ cells will be likely contaminated with CD56$^{bright}$ NK cells (*Loyon et al., 2019*; *Salomé et al., 2019*; *Trabanelli et al., 2019*; *Yudanin et al., 2019*). In contrast, the present data support previous work showing that NK cells can be accurately separated from other ILCs by consideration of CD94/NKG2A expression (*Bernink et al., 2013*; *Mjösberg et al., 2011*; *Scoville et al., 2017*).

Almost all NK cell precursors described so far are characterized by expression of CD117 (cKIT), the receptor for stem cell factor (SCF), including CD34$^+$ early progenitors (stage 2) as well as more differentiated CD34$^-$ stages (*Freud et al., 2006*). Moreover, it was previously shown that CD117 is gradually downregulated during NK cell differentiation in secondary lymph nodes from early to late stages of NK cell development with mature CD16$^+$CD56$^{dim}$ NK cells being the only CD117$^-$ stage (*Freud et al., 2016*). The absence of CD117 on ILC1-like NKPs represents thus an unusual feature for NK cell progenitors and demonstrates that cKIT signaling is not required for triggering their inherent NK cell differentiation potential. The lack of CD117 sets ILC1-like NKPs also apart from the recently described circulating Lin$^-$CD7$^+$CD127$^+$CD117$^+$ multipotent ILC progenitors (ILCp) that gave rise to all ILC subsets including NK cells (*Lim et al., 2017*). We thus hypothesize that circulating ILC1-like NKPs are at a more advanced developmental stage than CD34$^+$CD117$^+$ or

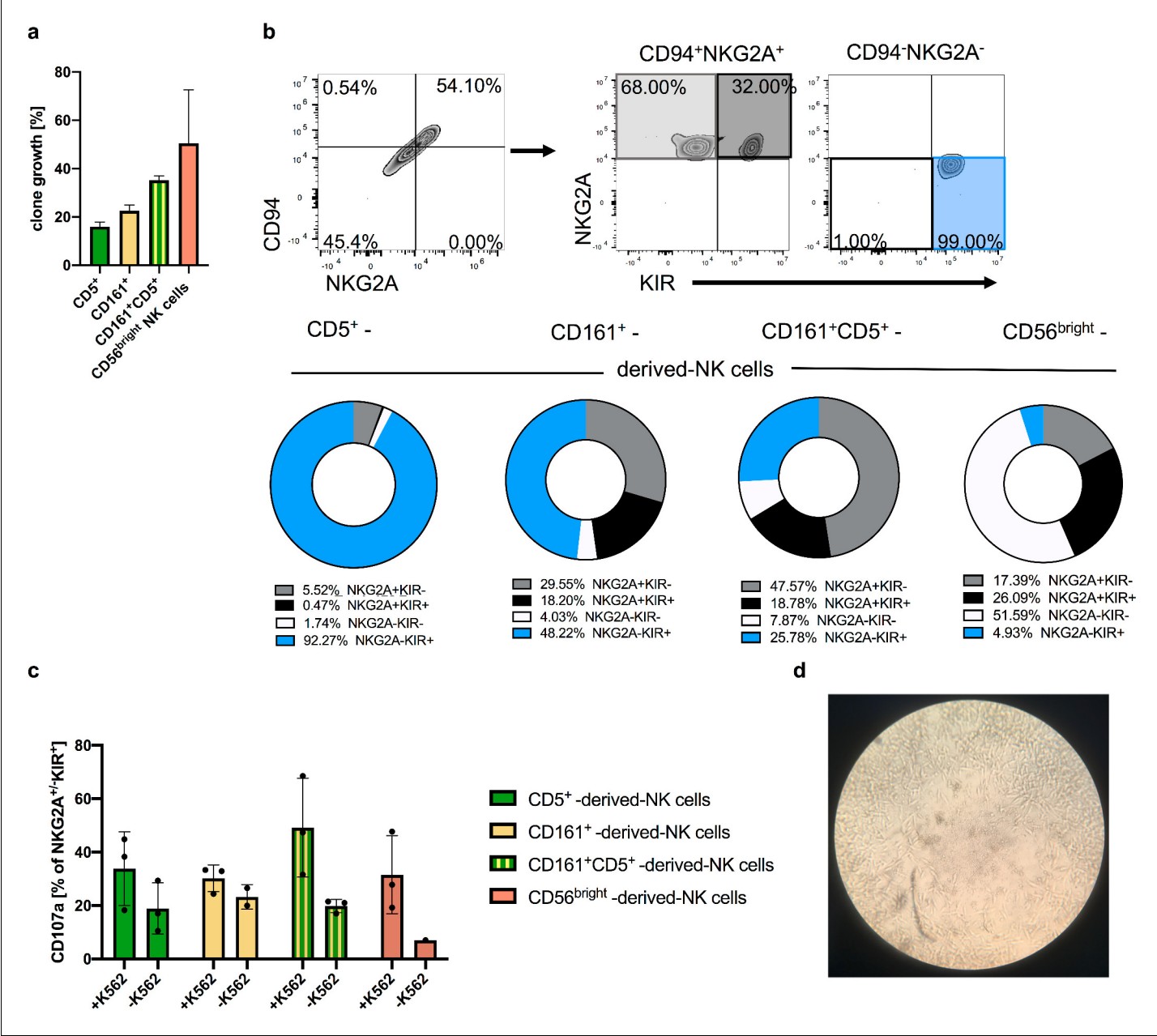

**Figure 7.** Clonal analyses of ILC1-like cells reveal high frequency of NKPs developing into cytotoxic KIR+NKG2A- NK cells. Single cells from the four ILC1-like subsets CD5+, CD161+, and CD161+CD5+ were flow cytometrically deposited in 96 well plates for clonal differentiation cultures on OP9-DL1 stroma cells. (a) Efficiency of clone growth at day 14. (b) Exemplary gating strategy for in vitro differentiated clones at day 28: CD94+NKG2A+ cells as well as CD94-NKG2A- cells were further divided on the basis of their respective NKG2A and KIR expression (upper panel). Pie charts and corresponding frequency of clones for the four different subsets (bottom panel): NKG2A+KIR- (grey), NKG2A+KIR+ (black), NKG2A-KIR- (white), and NKG2A-KIR+(blue) (n = 20 for all ILC1-like subsets, n = 4 for CD56bright NK cells). (c) Quantification of CD107a cytotoxic mobilization assay with K562 cells in an effector/target ratio of 1:1 from single cell cultures (n = 3). (d) Representative microscopic picture from single cell culture exhibiting erasure of feeder cells by developing NK cells in the central region of the well. The heights of the bars represent the mean ± SEM. Levels of significance were calculated with a One-Way ANOVA with a multiple correction post-test (Kruskal-Wallis test). Data were generated from 288 CD5+ and CD161+ cells each, 177 CD5+CD161+ cells, and 12 CD56bright NK cells sorted from a single donor.

CD34-CD117+NK cell progenitors and also compared to ILCp. However, the upstream progenitor of ILC1-like NK cells is currently elusive. Although it is possible that ILC1-like NK cells are developing from previously described CD117+ NKPs, a linear developmental relationship between them appears

to be unlikely since the specific potential to generate NKG2A⁻KIR⁺ NK cells could so far not be detected when starting from the more immature CD117⁺ NKPs.

Notably, some of the most significant transcriptomic differences between ILC1-like cells and NK cells pertained to the ILC1-specific expression of genes classically attributed to the T cell lineage such as *CD5*, *CD6*, *CD28*, and also components of the CD3/TCR complex, but without detectable expression of CD3 or TCR on the cell surface. The expression of T cell lineage-related molecules is in line with previous reports of ILC1 in peripheral blood (*Roan et al., 2016*) and tonsils by single cell RNAseq analysis (*Björklund et al., 2016*). Of note, in the course of our experiments we did not observe any potential for T cell development neither with OP9 nor OP9-DL1 and this was true in the presence of NK cell (IL-2, IL-7, and IL-15) as well as T cell (IL-7 and FLt3L) conditions. This contrasts to the properties of CD34⁺ early hematopoietic progenitors as reported previously (*Zúñiga-Pflücker, 2004*). Nonetheless, the T cell signature could be an indication of a thymic origin of the circulating ILC1-like NKPs. Of note, it has been proposed that NKPs develop in the thymus from progenitors with failed T-cell program (*Klein Wolterink et al., 2010*). Unlike the original model where NK/T bi-potent precursors directly convert to NK cells in the thymus after an unsuccessful attempt to become T-cells (*Klein Wolterink et al., 2010*), it could be speculated that 'T-cell failure' might lead to the release of CD5⁺ ILC1-like NKP from the thymus into the periphery. Our data suggested some evidence, as CB ILC1-like cells express either CD4 or CD8, intracellular CD3δ, CD2 was selectively expressed on CD5⁺ ILC1-like cells, and RNAseq analyses revealed a polyclonal repertoire of the genes *TRAV* and *TRBV* encoding the variable region of the T cell receptor alpha and beta chain, respectively. Of note, CD5⁺ ILC-like cells expressing intracellular CD3 components were previously isolated from the thymus (*Nagasawa et al., 2017*) and intracellular CD3δ expression was reported previously within human fetal NK cells (*Phillips et al., 1992*). Further studies of thymic ILC/NK progenitors will be necessary to better understand the origin of ILC1-like NKPs.

The prevailing view of NK cell development sees a linear consecutive relationship between CD117⁺ NKPs, CD56^bright, and finally CD56^dim NK cells (*Freud et al., 2006*). This model was majorly defined by analysis of progenitors in SLNs (*Freud et al., 2006*). The present data suggest that in the circulation, CD56^bright and CD56^dim NK cells might not be necessarily connected in a similar linear developmental relationship. Generally, CD56^bright NK cells mostly rely on NKG2A as their only and rather broad HLA class I-specific inhibitory receptor, whereas the more mature CD56^dim NK cells are characterized by complex HLA class I-educated KIR repertoires. The latter is necessary for sensitive detection of pathogen- or tumor-mediated downregulation of selected HLA class I genes. So far, a progenitor/product relationship has been widely assumed for CD56^bright and CD56^dim NK cells respectively, although other possibilities such as a branched model for NK cell development were suggested (*Cichocki et al., 2019*; *Michel et al., 2016*). Here, employing identical differentiation conditions, ILC1-like NKPs rather than CD56^bright NK cells were able to reconstitute complex NK cell receptor repertoires including NKG2A⁻KIR⁺ NK cells. On the other hand, CD56^bright NK cells showed a high proliferative capacity and mainly remained NKG2A⁺KIR⁻. Notably, cultured CD56^bright NK cells changed from expressing the CD56^bright-specific granzyme K to CD56^dim-specific granzyme B, indicative of the acquisition of effector functions and consequently were cytotoxic after 14d of culture. The data are thus compatible with a non-linear, 'branched' model in which CD56^bright NK cell differentiation preferentially leads to NKG2A⁺CD56^dim NK cells, whereas ILC1-like cells preferentially advance to the NKG2A⁻KIR⁺ stage, which would make the roles of CD56^bright and ILC1-like cells in building NK cell repertoires complimentary. In favor of this model, many of the T cell-specific markers that are in ILC1-like NKPs are also found in CD56^dim NK cells, whereas they are low or absent in CD56^bright NK cells. Examples of this are *TRBC1*, encoding the T cell receptor β constant region, *BCL11B* encoding a TF important for T cell function (*Hosokawa et al., 2020*), and *CD3D*.

The ILC1-like cells could be phenotypically and functionally further broken down into four subsets defined by expression of CD5 and CD161. While all subsets exhibited NK cell differentiation potential, they might differentially contribute to NK cell repertoire development.

When speculating on the developmental trajectories, CD5⁺CD161⁻ cells might represent the most immature subset. They are characterized by the expression of various T cell molecules such as CD2, CD6 and CD28, compatible with their recent exit from the thymus and also possess chemokine receptors for further migration into SLN for further differentiation. In contrast, CD5⁻CD161⁺ are likely to be a more differentiated ILC1-like subset since they have lost most of the T cell-specific molecules and chemokine receptors while acquiring effector functions. They might in fact remain in the

circulation as an independent SLN NK cell source. The CD5$^+$CD161$^+$ subset could constitute an intermediate subset as they possess T cell-specific molecules such as CD2 and chemokine receptors but have already acquire IFNγ effector function.

The frequency of ILC1-like cells was particularly high in CBs of early gestational age and decreased until term. Similar changes were not observed for other ILC groups that exhibited no changes in relation to gestational age but significantly decreased after birth in an ageing-like process. We hypothesize that the loss of ILC1-like cells is due to preferential migration into tissues during the perinatal phase of development. The large majority of ILC1-like cells expressed the chemokine receptor CCR7, which supports migration to SLN, an established site of NK cell maturation. Further subsets co-expressed CCR4 or CCR9, supporting migration to skin and the gastrointestinal tract. We thus suggest that the majority of ILC1-like NKPs travel to the SLNs before and around birth to differentiate into NK cells and eventually reenter circulation. The migration of ILC1-like NKPs to SLN would provide them with the necessary niche signals required for further maturation. A key signal required for NK cell maturation including expression of KIR is provided by NOTCH and the necessary NOTCH ligands are highly expressed in SLN (*Radtke et al., 2013*). Thus, the dependence of ILC1-like NKPs on NOTCH signaling as observed here in OP-9 differentiation assays would support the idea of migration to lymphatic tissue for successful final maturation.

In summary, the present work constitutes a thorough dissection of group 1 ILCs in neonatal blood on the molecular and functional level. We demonstrate that ILC1-like cells are very different from NK cells on the transcriptional, epigenetic, and functional level but instead constitute a potent NKP. The lin$^-$CD127$^+$CD117$^-$ ILC1-like NKP is distinguished from previously defined NKPs and ILCPs by the absence of CD117 and the presence of T cell-specific molecules and also by the property of generating diversified NK cell repertoires characterized by KIR expression as well as the downregulation of NKG2A. ILC1-like NKPs were found in CB and PB, but the latter were less potent in generating mature NK cells. The study suggests high spatial and temporal dynamics within group I ILC during perinatal development that is driven by the migratory properties of ILC1-like NKP. For realization of the NKP potential, ILC1-like NKPs would travel to SLN, where they are exposed to signals such as NOTCH ligands and IL-15 that induce NK cell maturation. Although the fate of ILC1 NKPs in SLN is unknown, we suggest that they are released back to the circulation for building up diversified NK cell repertoires after birth.

## Materials and methods

### Human samples and ethics statement

Buffy coats of anonymous, healthy blood donations were kindly provided by the Blutspendezentrale at the University Hospital Düsseldorf. Umbilical cord bloods used within this study were collected from the José Carreras Stem Cell Bank at the ITZ. The protocol used was accepted by the institutional review board at the University of Düsseldorf (study number 2019–383) and is in accordance to the Declaration of Helsinki. Blood samples were either processed directly or left at room temperature (RT) overnight and were processed the following day. Information about the week of gestation was provided by the mothers.

### Isolation of MNCs from cord blood and buffy coats

From each blood sample, aliquots were taken for KIR genotyping and assessment of whole blood cell count (Cell Dyn 3500R, Abbot Laboratories, Illinois, USA). CB (1:1) and buffy coats (1:2) were diluted with sterile 1xPBS (Gibco by Life Technologies, California, USA) and MNCs were isolated by density gradient centrifugation (Biocoll, 1.077 g/cm$^3$/ Biochrom Merck Millipore). Cells were resuspended in 5 ml of ice-cold ammonium chloride solution (pH = 7.4, University Clinic Düsseldorf) for 5 min at RT to lyse residual erythrocytes and washed three times afterwards. MNCs were counted and cryopreserved or directly used for further analyses.

### Flow cytometry analyses

Cells were extracellularly stained with the following FITC conjugated antibodies for the lineage panel, as previously described (*Bianca Bennstein et al., 2019*): anti-CD3 (UCHT1), anti-CD1a (HI149), anti-CD14 (HCD14), anti-CD19 (HIB19) anti-TCRαβ (IP26), anti-TCRγδ (B1), anti-CD123

(6H6), anti-CD303/BDCA-2 (201A), anti-FcεR1a (AER-37(CRA-1)), anti-CD235α (HI264), anti-CD66b (G10F5), anti-CD34 (581) all from BioLegend. Of note, anti-TCRαβ (clone: IP26), anti-TCRγδ (clone: B1) were not included when analyzing the age-related decline in CB, adult PB, and elderly adult PB. The following antibodies were further used within this study: anti-CD94-PE/Cy7 or -APC (DX22); anti- CD3 Brilliant violet (BV) 510 (UCHT1); anti-CD56-APC/Cy7, BV650 or PE/Dazzle 594 (HCD56); anti-CD117-PE or BV421 (104D2); anti-CRTH2-PE/Dazzle 594 (BM16); anti-CD161-Alexa Flour 700 (HP-3G10); anti-CD5-APC/Cy7 (L17F12); anti-CD6-PE (BL-CD6); anti-CD158b1,b2,j (2DL2/L3/S2)-FITC or -PE (DX27); CD158e1 (KIR3DL1) –Alexa Fluor 700 or –PE (DX9); CD158a,h,g (KIR2DL1/S1/S3/S5) –FITC or –PE (HP-MA4); anti-IFNγ-PE/Cy7 (B27); anti-CCR4-APC (L291H4), anti-CD107a-FITC (H4A3), and goat anti-mouse IgG (Poly4053), all from BioLegend (California, USA), anti-CD127-PE/Cy5 (R34.34), anti-CD28-PE (CD28.2), anti-NKG2A-APC (Z199), all from Beckman Coulter (California, USA), anti-CD158b2 (KIR2DL3)-FITC (180701) and anti-CCR9 unconjugated (248621), both from R and D systems. Anti-CCR7-PE-CF$^{594}$ (150503) was purchased by BD Bioscience (California, USA). Intranuclear staining of anti-Tbet-BV605 (4B10, BioLegend), and anti-Eomes- PE-eFluor610 (WD1928, Invitrogen) was performed with the FoxP3 staining kit (Thermo Fischer Scientific) and corresponding protocol. All flow cytometric analyses were performed on a Cytoflex (Beckman Coulter) with previously described settings (*Bianca Bennstein et al., 2019*). Analyses were performed on the Kaluza software 2.1 (Beckman Coulter). t-Distributed Stochastic Neighbor Embedding (t-SNE) analyses for chemokine receptor expression on CB ILC1-like cells was done employing a 10-color staining protocol. The cell surface receptors CD127, CCR7, CD161, CD5, with additional staining of CCR9 or CCR4 were used to apply t-SNE analyses for ILC1-like cells with 500 iterations using FlowJo software (BD). For further details on reagents used see Appendix 1—key resources table.

## Cell sorting

Monocytes, T- and B-cells were depleted in CB MNCs via MojoSort Streptavidin Nanobeads (BioLegend) using the supplier's negative selection protocol. In brief, ~10–20×10$^7$ cells were stained with the biotinylated antibodies anti-CD3 (OKT3, 3.2 µl/10 × 10$^7$ cells), anti-CD14 (63D3, 4.8 µl/10 × 10$^7$), anti-CD19 (HIB19, 4.8 µl/10 × 10$^7$), and anti-CD66b (G10F5, 2.4 µl/10 × 10$^7$) from BioLegend for 15 min on ice. After washing, the cells were incubated for 15 min on ice with MojoSort Streptavidin Nanobeads (50 µl/10 × 10$^7$, BioLegend). After an additional washing step, the cells were separated on a MOJO magnet for 5 min, harvested, and further stained with lineage panel and ILC inclusion antibodies for sorting. CD56$^{dim}$ NK cells (lin$^-$CD94$^+$CD56$^{dim}$), CD56$^{bright}$ NK cells (lin$^-$CD94$^+$CD56$^{bright}$), and ILC1-like cells (lin$^-$CD94$^-$CD127$^+$CD117$^-$CRTH2$^-$) were sorted for RNAseq analyses and the latter two for differentiation on OP9-DL1. Additionally, ILC2 (lin$^-$CD94$^-$CD127$^+$-CD117$^{-/+}$CRTH2$^+$) were sorted for in vitro differentiation on OP9-DL1. ILC1-like cells were further gated on CD5$^+$ and CD5$^-$. Cell sorting was performed on a MoFlo XDP (Beckman Coulter).

## Functional analyses

CB MNCs were stimulated in a 24-well plate with human (h) IL-12 (5 ng/ml) and IL-18 (50 ng/ml) overnight or 2 hr with PMA/Ionomycin at a concentration of 10 ng/ml PMA and 1 µg/ml Ionomycin. Subsequently, Brefeldin A Solution (1000X, BioLegend) was added to each well (dilution 1000–3000-fold depending on the length of the experiment) for an additional 2 hr (PMA/ionomycin) or 4 hr (cytokine stimulation). The cells were stained extracellularly to detect ILC surface markers and intracellularly for IFNγ expression using the intranuclear staining kit (Biolegend). Sorted ILC1-like and NK cells were stimulated with human (h) IL-12 (5 ng/ml) and hIL-18 (50 ng/ml) for 5 days, supernatant was collected at day 1 and 5 for LEGENDplex Human T Helper Cytokine Panel (BioLegend).

## Maintenance of cell lines

Stroma cell lines OP9 and OP9-DL1 were kindly provided by Prof. Dr. Zúñiga-Pflücker, University of Toronto and were cultivated in DMEM low glucose (1 g/l) + Glutamax (Gibco) with 1% Penicillin/Streptavidin (Gibco) and 10% Fetal calf serum (FCS) (Merck). Both cell lines were cultivated in T75 culture flasks, kept at 37°C with 5% CO$_2$, trypsinized, and split twice a week 1:5. K562 cells were cultured in DMEM high glucose (4,5 g/l) (Gibco) with 50 µg/ml Gentamycin (Gibco) and 10% FCS. Both cell lines were negatively tested for mycoplasma. The OP9-DL1 were tested by flow cytometry for DL1 expression (cells were GFP positive).

## Co-cultivation of OP9 and OP9-DL1 with PB or CB MNCs

5.000–50.000 OP9/OP9-DL1 cells per well were transferred into 96-flat bottom plates one day prior to sorting. Cells were isolated from CB, stained, and sorted as illustrated in *Figure 1—figure supplement 1*. After sorting, 1000–2500 ILC1-like, ILC2, or NK cells were added onto the cell layer and cultivated in' NK2 'medium (2/3 parts DMEM (4,5 g/ml glucose), 1/3 parts Ham's F12 (Biochrom), 10% human AB serum, 20 mg/L Ascorbic acid, 50µmol/L ethanolamine, 50 µg/L sodium-selenite (all from Sigma Aldrich), 24 µmol/L 2-mercaptoethanol, 1% Penicillin/Streptavidin, and 1% L-glutamine (all from Gibco) containing IL-2 (500 U/ml, Novartis), IL-7 (10 ng/ml), and IL-15 (5 ng/ml, both from Miltenyi Biotec, Germany) (*Figure 4a*). Medium was replaced every 3–5 days by removing half the volume of old medium and adding half the volume of fresh medium. The co-culture was phenotypically analyzed by flow cytometry after 8 and 14 days, functionality was assessed after 15–16 days by analyzing cytotoxic potential (CFSE killing assay) and degranulation (CD107a), respectively.

For single cell cloning, 3000 OP9-DL1 cells were transferred into 96 U bottom plates one day prior to sorting. Fresh medium was added twice a week by replacing half of the volume. Cloning efficiency was established at day 14 by microscopic inspection of each well individually.

## NK cell degranulation and killing assay

In vitro-cultivated ILCs and CD56$^{bright}$ NK cells were filtered through a 30 µm strainer to remove OP9-DL1 cells. For the CD107a degranulation assay, cells were incubated with K562 at a 1:1 E/T ratio with addition of a CD107a FITC mAb (H4A3, BioLegend) in a 96- round bottom plate with a centrifugation step of 5 min at 500 rpm. After 1 hr incubation at 37°C, 5% CO$_2$, Brefeldin A (1000fold dilution) and Monensin (20 nM, BioLegend) were added. After additional 4 hr, the cells were harvested, stained and analyzed via flow cytometry. As controls for the CD107a degranulation assay both cell types were seeded into a well without target cells, the spontaneous CD107a degranulation was subtracted from the target-specific CD107a degranulation.

For the CFSE assay, K562 cells were labelled with 5 mM CFSE for 10 min at 37°C, after two washings steps with PBS containing 20% FCS, the cells were incubated for 5 min and added in a ratio of 1:1 to in vitro generated ILC1-like- and CD56$^{bright}$-derived NK cells in a 96-round bottom plate. Unlabeled and CFSE-labelled K562 without effector cells served as controls. The plate was centrifuged for 5 min at 500 rpm. After 5 hr incubation at 37°C and 5% CO$_2$, cells were washed once with 1xPBS containing 0.5% BSA (Roth) and 5 mM EDTA (Roth). Shortly before flow cytometric analyses, 3 µl Propidium Iodide (PI, BioLegend) was added to each tube. Target-specific killing was calculated by subtracting the PI-positive fraction of the CFSE-labelled target cells without effector cells from the PI-positive fraction of the 1:1 mixture of effector: target cells. Target-specific killing = % of CFSE$^+$PI$^+$ target cells with effector cells 1:1 % of CFSE$^+$PI$^+$ target cells alone.

For the ADCC assay CD20$^+$ Raji and Rituximab (anti-CD20) were used. ILC1-like and CD56$^{bright}$-derived NK cells were pre-incubated with 1 µg/ml Rituximab for 10 min at 37°C, 5% CO$_2$ before adding Raji as target cells. An E/T ratio of 1:1 was used and the same protocol as for the CD107a degranulation assay was used (see above). As controls, ILC1-like and CD56$^{bright}$ -derived NK cells were mixed at an E/T ratio of 1:1 with Raji without Rituximab. All cell lines were negatively tested for mycoplasma.

## KIR genotyping

KIR genotyping was carried out by sequence-specific primer-polymerase chain reaction (PCR-SSP), as previously described (*Uhrberg et al., 2002*).

## ATAC sequencing

ATACseq (assay for transposase-accessible chromatin using sequencing) was performed by sorting 5000 ILC1-like cells and NK cells (see *Figure 1—figure supplement 1* for gating). The cells were centrifuged at 500xg for 5 min at 4°C. The transposase reaction mix (2x transposase buffer, TDE1 enzyme (both from Illumina), 0.01% Digitonin (Promega)) was incubated for 30 min at 37°C with a rotation of 300 rpm (*Corces et al., 2016*). The DNA was isolated using the Elute clean up kit according to manufacturer's protocol (Qiagen). The processed DNA was amplified and run on an Illumina HiSeq4000 instrument (paired-end 2 × 100 bp). Adapter detection was done using detect_adapter. py of the ENCODE-ATACseq-pipeline (https://github.com/kundajelab/atac_dnase_

pipelines; *Lee et al., 2014*). Detected adaptors were finally trimmed using cutadapt (version 2.3; *Martin, 2011*). The results were mapped against the human genome (GRCh38, released 2014) using bowtie2 (version 2.3.4.; *Langmead and Salzberg, 2012*). Afterwards, multi-mapping reads, duplicates and reads mapping against the mitochondrial DNA were detected and removed using PICARD (version 2.20.2; http://broadinstitute.github.io/picard/) and SAMtools (version 1.9; *Li et al., 2009*). Due to the observation that Tn5 transposase binds as a dimer and inserts two adaptors separated by 9 bp (*Adey et al., 2010*), a read-shifting-step was fulfilled using alignmentSieve of deepTools (*Ramírez et al., 2016*). Finally, peak-calling was done using macs2 (-f BAMPE -g hs –`keep-dup` all –`cutoff-analysis`; *Zhang et al., 2008*). To visualize the results, browser tracks for the Genome Browser of the University of California, Santa Cruz (UCSC; https://genome.ucsc.edu/) were created by converting BAM- to BIGWIG-files using bamCoverage of deepTools (*Ramírez et al., 2016*) and a normalization step of 1x effective genome size (https://deeptools.readthedocs.io/en/latest/content/feature/effectiveGenomeSize.html).

## RNA sequencing and data analysis

After cell sorting, cells were stored in TRIzol Reagent (Invitrogen) and total RNA was extracted and fragmented. Reverse transcription and library production were carried out in the NGS integrative Genomics (NIG) facility in Göttingen, Germany with an Illumina Truseq RNA preparation kit as described in the company's protocol. Sequencing of the libraries was performed with an Illumina HiSeq4000 (single-read 1 × 50 bp). Sequence reads were mapped to the human genome (hg38) with STAR (version STAR_2.5Oa) and read counts of gene transcripts were determined using gtf file Homo_sapiens.GRCH38.84.gtf and featureCount (v1.5.0-p1). Analysis of differential gene transcription and normalization of read counts and PCA were performed with R package DESeq2 (v.1.22.2) (*Love et al., 2014*). Four-way plots were generated with R package vidger (v.1.2.1) (*McDermaid et al., 2019*).

## Statistical analyses

All tests were performed with a parametric or nonparametric assumption (depending on normal distribution) and a 0.05 significance level. All analyses were done using the GraphPad Prism 8.0.0 (GraphPad Software, San Diego, California USA, www.graphpad.com).

# Acknowledgements

The authors thank the CB and PB donors for providing blood samples. This work was supported by funds from the Düsseldorf School of Oncology (funded by the Comprehensive Cancer Center Düsseldorf/Deutsche Krebshilfe and the Medical Faculty HHU Düsseldorf) and the Deutsche Forschungsgemeinschaft DFG SPP1937-UH91/8-1 (MU). The authors declare no competing financial interests.

# Additional information

### Funding

| Funder | Grant reference number | Author |
|---|---|---|
| Deutsche Forschungsgemeinschaft | DFG SPP1937-UH91/8-1 | Markus Uhrberg |

The funders had no role in study design, data collection and interpretation, or the decision to submit the work for publication.

### Author contributions

Sabrina Bianca Bennstein, Conceptualization, Data curation, Formal analysis, Validation, Visualization, Methodology, Writing - original draft, Project administration, Writing - review and editing; Sandra Weinhold, Data curation, Formal analysis; Angela Riccarda Manser, Supervision, Methodology; Nadine Scherenschlich, Data curation, Validation, Methodology; Angela Noll, Formal analysis, Methodology; Katharina Raba, Data curation, Methodology; Gesine Kögler, Resources; Lutz Walter, Formal analysis, Validation, Visualization, Methodology; Markus Uhrberg, Conceptualization, Resources,

Supervision, Funding acquisition, Investigation, Writing - original draft, Project administration, Writing - review and editing

## Author ORCIDs
Sabrina Bianca Bennstein (iD) https://orcid.org/0000-0003-0477-2748
Markus Uhrberg (iD) https://orcid.org/0000-0001-9553-1987

## Ethics

Human subjects: Buffy coats of anonymous, healthy blood donations were kindly provided by the Blutspendezentrale at the University Hospital Düsseldorf. Umbilical cord bloods used within this study were collected from the José Carreras Stem Cell Bank at the ITZ. The protocol used was accepted by the institutional review board at the University of Düsseldorf (study number 2019-383) and is in accordance to the Declaration of Helsinki.

## Decision letter and Author response

Decision letter https://doi.org/10.7554/eLife.55232.sa1
Author response https://doi.org/10.7554/eLife.55232.sa2

# Additional files

## Supplementary files

• Transparent reporting form

## Data availability

RNA sequencing and ATACseq data is accessible at NCBI Project ID: PRJNA594493 (http://www.ncbi.nlm.nih.gov/bioproject/594493).

The following dataset was generated:

| Author(s) | Year | Dataset title | Dataset URL | Database and Identifier |
|---|---|---|---|---|
| Walter L, Uhrberg M | 2019 | Identification of an ILC1-like NK cell progenitor in neonatal blood | https://www.ncbi.nlm.nih.gov/bioproject/594493 | NCBI BioProject, PRJNA594493 |

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

# Appendix 1

## Appendix 1—key resources table

| Reagent type (species) or resource | Designation | Source or reference | Identifiers | Additional information |
|---|---|---|---|---|
| Cell line (*M. musculus*) | OP9-DL1 | Schmitt, T. M. & Zúñiga-Pflücker, J. C. Induction of T cell development from hematopoietic pro-genitor cells by delta-like-1 in vitro. *Immunity* 17, 749–756 (2002). | No identifier, but similar to RCB Cat# RCB2927, RRID:CVCL_B220 | Provided by Prof. Dr. Zúñiga-Pflücker, University of Toronto |
| Cell line (*M. musculus*) | OP9 | Kodama, H., Nose, M., Niida, S. & Nishikawa, S. Involvement of the c-kit receptor in the adhesion of hematopoietic stem cells to stromal cells. Exp. Hematol. 22, 979–984 (1994). | No identifier, but similar to RCB Cat# RCB2926, RRID:CVCL_B219 | Provided by Prof. Dr. Zúñiga-Pflücker, University of Toron-to |
| Cell line (*H. sapiens*) | Raji | ATCC | ATCC Cat# CRL-7936, RRID:CVCL_0511 | Cell line maintained in lab |
| Cell line (*H. sapiens*) | K562 | ATCC | ATCC Cat# CCL-243, RRID: CVCL_0004 | Cell line maintained in lab |
| Biological sample (*H. sapiens*) | ILC1-like cells; CB ILC1-like cells | Freshly isolated from do-nated umbilical cord blood (CB) within the lab | | José Carreras Stem Cell Bank at the ITZ, University Hospital Düsseldorf |
| Biological sample (*H. sapiens*) | ILC2; CB ILC2 | Freshly isolated from do-nated umbilical cord blood (CB) within the lab | | José Carreras Stem Cell Bank at the ITZ, University Hospital Düsseldorf |
| Biological sample (*H. sapiens*) | CD56$^{bright}$ NK cells; CB CD56$^{bright}$ NK cells | Freshly isolated from do-nated umbilical cord blood (CB) within the lab | | José Carreras Stem Cell Bank at the ITZ, University Hospital Düsseldorf |
| Biological sample (*H. sapiens*) | CB MNCs (mono nu-clear celLs) | Freshly isolated from do-nated umbilical cord blood (CB) within the lab | | José Carreras Stem Cell Bank at the ITZ, University Hospital Düsseldorf |
| Biological sample (*H. sapiens*) | ILC1-like; PB ILC1-like cells | Freshly isolated from healthy blood donors in the lab | | Blutspendezentrale at the University Hospital Düsseldorf |
| Biological sample (*H. sapiens*) | CD56$^{bright}$ NK cells; PB CD56$^{bright}$ NK cells | Freshly isolated from healthy blood donors in the lab | | Blutspendezentrale at the University Hospital Düsseldorf |
| Antibody | anti- CD3-FITC (Mouse monoclonal) | Biolegend | BioLegend Cat# 300405, RRID:AB_314059 | "1:100" |
| Antibody | anti- CD3- BV510 (Mouse monoclonal) | Biolegend | BioLegend Cat# 300447, RRID:AB_2563467 | "1:200" |

*Appendix 1—key resources table continued*

| Reagent type (species) or resource | Designation | Source or reference | Identifiers | Additional information |
|---|---|---|---|---|
| Antibody | anti-CD1a -FITC (Mouse monoclonal) | Biolegend | BioLegend Cat# 300104, RRID:AB_314018 | "1:25" |
| Antibody | anti-CD14- FITC (Mouse monoclonal) | Biolegend | BioLegend Cat# 325604, RRID:AB_830677 | "1:100" |
| Antibody | anti-CD19 -FITC (Mouse monoclonal) | Biolegend | BioLegend Cat# 302206, RRID:AB_314236 | "1:200" |
| Antibody | anti-TCR$\alpha\beta$-FITC (Mouse monoclonal) | Biolegend | BioLegend Cat# 306705, RRID:AB_314639 | "1:100" |
| Antibody | anti-TCR$\gamma\delta$ -FITC (Mouse monoclonal) | Biolegend | BioLegend Cat# 331207, RRID:AB_1575111 | "1:12.5" |
| Antibody | anti-CD123 -FITC (Mouse monoclonal) | Biolegend | BioLegend Cat# 306014, RRID:AB_2124259 | "1:200" |
| Antibody | anti-CD303 (BDCA-2) -FITC (Mouse monoclonal) | Biolegend | BioLegend Cat# 354208, RRID:AB_2561364 | "1:25" |
| Antibody | anti- Fc$\varepsilon$R1$\alpha$ -FITC (Mouse monoclonal) | Biolegend | BioLegend Cat# 334608, RRID:AB_1227653 | "1:25" |
| Antibody | anti-CD235$\alpha$ -FITC (Mouse monoclonal) | Biolegend | BioLegend Cat# 349104, RRID:AB_10613463 | "1:100" |
| Antibody | anti-CD66b -FITC (Mouse monoclonal) | Biolegend | BioLegend Cat# 305104, RRID:AB_314496 | "1:100" |
| Antibody | anti-CD34 -FITC (Mouse monoclonal) | Biolegend | BioLegend Cat# 343504, RRID:AB_1731852 | "1:100" |
| Antibody | anti-CD94 -APC (Mouse monoclonal) | Biolegend | BioLegend Cat# 305508, RRID:AB_2133129 | "1:100" |
| Antibody | anti-CD94- PE/Cy7 (Mouse monoclonal) | Biolegend | BioLegend Cat# 305516, RRID:AB_2632753 | "1:100" |
| Antibody | anti-CD56-APC/Cy7 (Mouse monoclonal) | Biolegend | BioLegend Cat# 318332, RRID:AB_10896424 | "1:100" |

*Appendix 1—key resources table continued*

| Reagent type (species) or resource | Designation | Source or reference | Identifiers | Additional information |
|---|---|---|---|---|
| Antibody | anti-CD56- BV650 (Mouse monoclonal) | Biolegend | BioLegend Cat# 318344, RRID:AB_2563838 | "1:100" |
| Antibody | anti-CD56-PE/Dazzle594 (Mouse monoclonal) | Biolegend | BioLegend Cat# 318348, RRID:AB_2563564 | "1:100" |
| Antibody | anti-CD117-PE (Mouse monoclonal) | Biolegend | BioLegend Cat# 313204, RRID:AB_314983 | "1:100" |
| Antibody | anti-CD117 BV421 (Mouse monoclonal) | Biolegend | BioLegend Cat# 313216, RRID:AB_11148721 | "1:100" |
| Antibody | anti-CRTH2 -PE/Dazzle 594 (Rat monoclonal) | Biolegend | BioLegend Cat# 350126, RRID:AB_2572053 | "1:40" |
| Antibody | anti-CD161 -Alexa Flour (Mouse monoclonal) | Biolegend | BioLegend Cat# 339942, RRID:AB_2565870 | "1:25" |
| Antibody | anti-CD5 -APC/Cy7 (Mouse monoclonal) | Biolegend | BioLegend Cat# 364010, RRID:AB_2564506 | "1:100" |
| Antibody | anti-CD6 -PE (Mouse monoclonal) | Biolegend | BioLegend Cat# 313906, RRID:AB_2260227 | "1:100" |
| Antibody | anti-CD158b1,b2,j; KIR2DL2/L3/S2-FITC (Mouse monoclonal) | Biolegend | BioLegend Cat# 312604, RRID:AB_2296486 | "1:100" |
| Antibody | anti-CD158b1,b2,j; KIR2DL2/L3/S2-PE (Mouse monoclonal) | Biolegend | BioLegend Cat# 312606, RRID:AB_2130554 | "1:100" |
| Antibody | anti-CD158e1; KIR3DL1 - Alexa Flour (Mouse monoclonal) | Biolegend | BioLegend Cat# 312712, RRID:AB_2130824 | "1:400" |
| Antibody | anti-CD158e1; KIR3DL1 – PE (Mouse monoclonal) | Biolegend | BioLegend Cat# 312708, RRID:AB_2249498 | "1:100" |
| Antibody | anti- CD158a,h,g; KIR2DL1/S1/S3/S5 - FITC (Mouse monoclonal) | Biolegend | BioLegend Cat# 339504, RRID:AB_2130378 | "1:100" |
| Antibody | anti- CD158a,h,g; KIR2DL1/S1/S3/S5 - PE (Mouse monoclonal) | Biolegend | BioLegend Cat# 339506, RRID:AB_2130374 | "1:100" |

*Appendix 1—key resources table continued*

| Reagent type (species) or resource | Designation | Source or reference | Identifiers | Additional information |
|---|---|---|---|---|
| Antibody | anti-IFN$\gamma$ -PE/Cy7 (Mouse monoclonal) | Biolegend | BioLegend Cat# 506518, RRID:AB_2123321 | "1:20" |
| Antibody | anti-CCR4 -APC (Mouse monoclonal) | Biolegend | BioLegend Cat# 359408, RRID:AB_2562429 | "1:100" |
| Antibody | anti-CD107a -FITC (Mouse monoclonal) | Biolegend | BioLegend Cat# 328606, RRID:AB_1186036 | "1:200" |
| Antibody | anti-Mouse -PE (Goat polyclonal) | Biolegend | BioLegend Cat# 405307, RRID:AB_315010 | "1:250" |
| Antibody | anti-CD127 -PE/Cy5 (Mouse monoclonal) | Beckman Coulter | Beckman Coulter Cat# A64617, RRID:AB_2833010 | "1:40" |
| Antibody | anti-CD28 -PE (Mouse monoclonal) | Beckman Coulter | Beckman Coulter Cat#I-M2071U, RRID:AB_2833011 | "1:10" |
| Antibody | anti-NKG2A -APC (Mouse monoclonal) | Beckman Coulter | Beckman Coulter Cat# A60797, RRID:AB_10643105 (conjugate APC) | "1:100" |
| Antibody | anti-CD158b2; KIR2DL3 -FITC (Mouse monoclonal) | R&D | R and D Systems Cat# FAB2014F-100, RRID:AB_2833013 | "1:20" |
| Antibody | anti-CCR9 (Mouse monoclonal) | R&D | R and D Systems Cat# MAB1791, RRID:AB_2073268 | "1:100" caution only stable for 4 month |
| Antibody | anti- CCR7 -PE-CF[594] (Mouse monoclonal) | BD Bioscience | BD Biosciences Cat# 562381, RRID:AB_11153301 | "1:20" |
| Antibody | anti-Tbet -BV605 (Mouse monoclonal) | Biolegend | BioLegend Cat# 644817, RRID:AB_11219388 | "1:20" |
| Antibody | anti- Eomes-PE-eFlour610 (Mouse monoclonal) | Invitrogen | Thermo Fisher Scientific Cat# 61-4877-42, RRID:AB_2574616 | "1:20" |
| Antibody | anti-CD3 -biotin (Mouse monoclonal) | Biolegend | BioLegend Cat# 317320, RRID:AB_10916519 | 3.2µl/10x10$^7$ cells |

*Appendix 1—key resources table continued*

| Reagent type (species) or resource | Designation | Source or reference | Identifiers | Additional information |
|---|---|---|---|---|
| Antibody | anti-CD14 biotin (Mouse monoclonal) | Biolegend | BioLegend Cat# 367105, RRID:AB_2566617 | 4.8µl/10x10$^7$ |
| Antibody | anti-CD19 -biotin (Mouse monoclonal) | Biolegend | BioLegend Cat# 302204, RRID:AB_314234 | 4.8µl/10x10$^7$ |
| Antibody | anti-CD66b-biotin (Mouse monoclonal) | Biolegend | ioLegend Cat# 305120, RRID: AB_2566608 | 2.4µl/10x10$^7$ |
| Antibody | anti-CD8a -BV510 (Mouse monoclonal) | Biolegend | BioLegend Cat# 301048, RRID:AB_2561942 | "1:50" |
| Antibodx | anti-CD4-APC (Mouse monoclonal) | Biolegend | BioLegend Cat# 317416, RRID:AB_571945 | "1:100" |
| Antibody | anti-CD3delta -R-PE (Mouse monoclonal) | *life*technologies | Thermo Fisher Scientific Cat# MHCD0304, RRID:AB_10376004 | "1:50" |
| Antibody | anti-CD2-APC (Mouse monoclonal) | Biolegend | BioLegend Cat# 300214, RRID:AB_10895925 | "1:100" |
| Commercial assay or kit | Foxp3 staining kit | ThermoFisher | Cat: 00-5523-00 | |
| Commercial assay or kit | Fixation buffer | Biolegend | Cat: 420801 | |
| Commercial assay or kit | LEGENDplex | Biolegend | Cat: 740722 | Human T Helper Cytokine Panel |
| Commercial assay or kit | MojoSortStreptavidin Nanobeads | Biolegend | Cat: 480016 | 50µl/10x10$^7$ |
| Commercial assay or kit | Illumina Tagment DNA Enzyme and Buffer Small Kit | Illumina | Cat: 15027865 | TDE1 enzyme |
| Software, algorithm | Kaluza 2.1 | Beckman Coulter | Kaluza, RRID:SCR_016182 | Version 2.1 |
| Software, algorithm | t-SNE embedded in FlowJo | BD Bioscience | FlowJo, RRID:SCR_008520 | 500 iterations for the t-SNE calculations |
| Software, algorithm | ENCODE-ATACseq-pipeline | https://github.com/kundajelab/atac_dnase_pipelines | | |
| Software, algorithm | cutadapt | *Martin, 2011* | cutadapt, RRID:SCR_011841 | Version 2.3 |
| Software, algorithm | Bowtie2 | *Langmead and Salzberg, 2012* | | Version 2.3.4 |
| Software, algorithm | PICARD | http://broadinstitute.github.io/picard/ | Picard, RRID:SCR_006525 | Version 2.20.2 |

*Appendix 1—key resources table continued*

| Reagent type (species) or resource | Designation | Source or reference | Identifiers | Additional information |
|---|---|---|---|---|
| Software, algorithm | SAMtools | *Li et al., 2009* | SAMTOOLS, RRID:SCR_002105 | Version 1.9 |
| Software, algorithm | Deeptools bamCoverage | *Ramírez et al., 2016* | Deeptools, RRID:SCR_016366 | |
| Software, algorithm | Macs2 | *Zhang et al., 2008* | | |
| Software, algorithm | R package DESeq2 | *Love et al., 2014* | DESeq2, RRID:SCR_015687 | Version 1.22.2 |
| Software, algorithm | R package vidger | MacDermaid et al 2018 | | Version 1.2.1 |
| Software, algorithm | GraphPad Prism | www.graphpad.com | GraphPad Prism, RRID:SCR_002798 | Version 8.0.0 |

