## [Decision Letter]

**Acceptance summary:**

In this study, Bennstein and colleagues investigated the relationship between ICL1 and NK cells in the cord blood. The authors show that cord blood ILC1 are a distinctive population of cells lacking signature NK cell genes and markers and instead expressing genes and markers of T-cell lineage. In appropriate culture conditions, a T-bet negative ILC1-like cell population can differentiate into a complex repertoire of functional NK cells that express both NKG2A and diverse KIRs. The conclusions are supported by the data and the results are relevant in the field of innate immune cell development.

**Decision letter after peer review:**

Thank you for submitting your article "Umbilical cord blood-derived ILC1-like cells constitute a novel precursor for mature KIR^+^NKG2A^-^ NK cells" for consideration by *eLife*. Your article has been reviewed by three peer reviewers, and the evaluation has been overseen by a Reviewing Editor and Tadatsugu Taniguchi as the Senior Editor. The following individuals involved in review of your submission have agreed to reveal their identity: Francesco Colucci (Reviewer #1); Nicholas Huntington (Reviewer #2).

The reviewers have discussed the reviews with one another and the Reviewing Editor has drafted this decision to help you prepare a revised submission.

In this study, Bennstein et al. describe a T-bet negative ILC1-like cell population identified in cord blood. These cells lack the signature of NK cell genes and markers, and instead express genes and markers of T-cell lineage. In appropriate culture conditions these cells can differentiate into a complex repertoire of functional NK cells that express both NKG2A and diverse KIRs. The reviewers appreciated the attention to an important topic, but raised substantial number of concerns about the manuscript as it currently stands. We therefore ask the authors to modify the manuscript according to the review recommendations before we can re-consider your manuscript for *eLife*.

Essential revisions:

1) The authors state that ILC1-like cells which are T-bet and CD56 negative and lack expression of perforin and all 5 granzymes develop into effector NK cells with de novo CD94, NKG2A and KIR2DL3 expression. The authors need to further phenotype the differentiated cells, showing evidence of essential NK cell markers, including CD56, NKp46, granzyme B and perforin. Additionally, they should demonstrate that inhibitory KIR expression in the differentiated cells is functionally inhibitory and leads to increased cytotoxicity in educated cells. Despite the ILC1-like derived NK cells having increased KIR expression compared to the CD56bright-derived NK cells, CD56bright-derived NK cells were equally functional and exhibited more target-specific degranulation compared to ILC1-derived NK cells.

2) The idea proposed in the discussion that CB ILC1 may be cells that have failed to convert into T cells into the thymus is an attractive one and perhaps the authors may wish to test it by looking at markers of recent thymic emigrants in CB ILC1 – if possible?

3) It is unclear if proper functional controls were utilized in this study. Target-specific degranulation needs to be shown instead of total degranulation in Figure 5, as Figure 7C makes evident. Additionally, the equation used to calculate cytotoxicity for the CFSE-based method should be included in the Materials and methods section. For the ADCC assay, it is unclear what cells were used for the control. Individual controls (antibody negative) for each cell population should be included.

4) How do the authors explain the phenotypic and functional differences between the individual ILC1-like subsets as defined by CD5 and CD161 expression and how do these individual subsets contribute to their proposed NKP potential? This point should be discussed in more detail.

5) The authors state that ILC1-like cells preferentially differentiate into mature KIR^+^ NK cells compared to CD56bright NK cells in the OP9-DL1 differentiation setting in the presence of IL-2, IL7 and IL-15. Cytokine stimulation (IL-2 and IL-15) of NK cells leading to the induction of proliferation and results in CD56 and NGK2A upregulation, even in mature CD56dim NK cells. Hence it is not surprising that CD56bright NK cells retained high NKG2A expression while actively proliferating. The present experimental setup therefore does not support their statement in paragraph five of the Discussion of a branched NK cell lineage model.

6) The authors clearly show that the CD127^+^ Lin- population is highly heterogenous, just by looking at CD161, CD5, CCR9, CCR4 and CCR7. Therefore, the transcriptomic and epigenetic data on the bulk population are not informative. Single cell analysis should be used to define the heterogeneity, considering that Simoni et al. have previously reported the heterogenous nature of human ILC1s (Simoni et al., Immunity, 2017) and questioned the nature of lineages included.

7) The authors claim that the cells do not generate T cells. However, they only use IL-7 and FLT3, while in other protocols the used IL-7, FLT3L and SCF.

[Editors' note: further revisions were suggested prior to acceptance, as described below.]

Thank you for submitting your revised manuscript "Umbilical cord blood-derived ILC1-like cells constitute a novel precursor for mature KIR^+^NKG2A^-^ NK cells" for *eLife*. Your revised manuscript was evaluated by three original reviewers, and the evaluation has been overseen by a Reviewing Editor and a Senior Editor. The reviewers acknowledged that in the revised version of the manuscript you successfully dealt with most of questions raised by reviewers, including additional experiments that were presented in the revised manuscript.

However, there is still one essential issue which you failed to address to comply with reviewer's recommendations. In our Decision Letter it was explicitly mentioned that "12. The ATAC-sequencing data is based on a single donor. Considering the large variation among individual donors, a sample size of 1 is not adequate to make conclusions". Therefore, unless you address this issues adequately, we cannot be committed to accept your paper for publication in *eLife*.

We would like to draw your attention to changes in our revision policy that we have made in response to COVID-19 (https://elifesciences.org/articles/57162). First, because many researchers have temporarily lost access to the labs, we will give authors as much time as they need to submit revised manuscripts. We are also offering, if you choose, to post the manuscript to bioRxiv (if it is not already there) along with this decision letter and a formal designation that the manuscript is "in revision at *eLife*". Please let us know if you would like to pursue this option. (If your work is more suitable for medRxiv, you will need to post the preprint yourself, as the mechanisms for us to do so are still in development.)

---

## [Author Response]

Essential revisions:1) The authors state that ILC1-like cells which are T-bet and CD56 negative and lack expression of perforin and all 5 granzymes develop into effector NK cells with de novo CD94, NKG2A and KIR2DL3 expression. The authors need to further phenotype the differentiated cells, showing evidence of essential NK cell markers, including CD56, NKp46, granzyme B and perforin. Additionally, they should demonstrate that inhibitory KIR expression in the differentiated cells is functionally inhibitory and leads to increased cytotoxicity in educated cells. Despite the ILC1-like derived NK cells having increased KIR expression compared to the CD56bright-derived NK cells, CD56bright-derived NK cells were equally functional and exhibited more target-specific degranulation compared to ILC1-derived NK cells.

As suggested by the reviewers, we have characterized the differentiated cells further by extracellular staining for CD56 and NKp46 and intracellular staining for perforin and granzyme B. As expected for effector NK cells, differentiated ILC1-like cells expressed CD56, NKp46 (low expression because of internalization in culture conditions), granzyme B, and perforin (Figure 4—figure supplement 2).

As pointed out by the reviewers the ILC1-derived NK cells exhibited functional features comparable to CD56^bright^ NK cells. This is not surprising since not only ILC1-like but also CD56^bright^ NK cells differentiated into effector NK cells within the 14d culture period. It is well described that CD56 is generally upregulated on all NK cells during culture, i.e. CD56^dim^ effector NK cell become CD56^bright^ after a few days and CD56^bright^ NK cells keep their strong expression of CD56 in culture, independent of other phenotypic and functional changes. The CD56^bright^ status is thus a kind of culture artefact. In order to demonstrate that the CD56^bright^ NK cells indeed converted to otherwise typical effector NK cells in culture, we additionally stained granzyme K, which is one of the few *bona fide* markers characterizing CD56^bright^ NK cells. As shown in the new Figure 4—figure supplement 2, ex vivo isolated CD56^bright^ cells express GzmK but not GzmB whereas CD56^dim^ expressed GzmB only. In our differentiation cultures CD56^bright^ NK cells loose GzmK and acquire GzmB expression, demonstrating their conversion to effector NK cells. (subsection “Neonatal ILC1-like cells contain a novel NK cell progenitor”, Figure 4—figure supplement 2)

We agree with the reviewers that it is an interesting question if the in vitro generated NK cells are licensed and if they exhibit a higher cytotoxic capacity. However, we are unable to address this question with the in vitro system we used. In order for NK cells to be licensed the respective KIR (2DL1/ 2DL2/L3) expressed on the NK cell needs to recognize the cognate self HLA-C (C2/C1). This is a process naturally happening in vivo, however our system generated NK cells on the murine feeder cell line OP9-DL1. Therefore, we are generating KIR-expressing NK cells in the absence of self HLA-C. Hence, they have not been educated due to the absence of self-ligands. This is the reason, why we could not address this question.

2) The idea proposed in the discussion that CB ILC1 may be cells that have failed to convert into T cells into the thymus is an attractive one and perhaps the authors may wish to test it by looking at markers of recent thymic emigrants in CB ILC1 – if possible?

Recent thymic emigrants are hard to characterize on the basis of a certain phenotype, as several publications observed different cell surface receptors depending on the T cell lineage of either CD8 or CD4(Fink, 2013; McFarland et al., 2000; Pekalski et al., 2017). Nonetheless, we were able to characterize our CB ILC1-like cells with regard to other T cell associated molecules in more detail and added a Supplementary Figure (Figure 1—figure supplement 3) and added information to the Results (paragraph two). Like recent thymic emigrants, CB ILC1-like cells either expressed CD4 or CD8 and intracellular CD3𝛿. We further detected specific CD2 expression within CD5^+^ ILC1-like cells. We also observed a diverse repertoire of the variable regions of the α and β T cell receptor chains (*TRAV* and *TRBV*) within our RNAseq data of sorted CB ILC1-like cells, suggesting that ILC1-like cells are polyclonal on the basis of rearranged TCR. The data further support the idea of a conversion of ILC1-like cells into NK cells after failing to become T cells within the thymus, but in the absence of more definitive proof it still remains a hypothesis.

3) It is unclear if proper functional controls were utilized in this study. Target-specific degranulation needs to be shown instead of total degranulation in Figure 5, as Figure 7C makes evident. Additionally, the equation used to calculate cytotoxicity for the CFSE-based method should be included in the Materials and methods section. For the ADCC assay, it is unclear what cells were used for the control. Individual controls (antibody negative) for each cell population should be included.

We thank the reviewers for highlighting this point and apologize if the respective statements within the manuscript were simplified. In fact, CD107a degranulation, CFSE, and ADCC assays were performed with proper controls. For the CD107a degranulation assay, we did already show target-specific degranulation, as we subtracted spontaneous CD107a degranulation (cells without target) from target specific CD107a degranulation (cells with K562). We have now introduced representative dot plots of spontaneous CD107a degranulation and the respective bar graphs (Figure 5A). We have also clarified this more clearly in the legend of Figure 5 as well as within the Material and methods section.

We have included a sentence and equation how we calculated the cytotoxicity for the CFSE based method.

Since in the ADCC experiment we originally included only a control of cells without target and Rituximab, we now added new data including controls with target cells but without antibodies as suggested by the reviewers. Similar to the CD107a degranulation assay, we newly included representative dot plots of target-specific killing (in vitro cultured cell with Raji as target and Rituximab) and spontaneous killing (in vitro cultured cell with Raji as target without Rituximab). Again, the ADCC data were comparable between ILC1-like and CD56^bright^ cells. Unfortunately, in case of ILC1-like derived NK cells the data did not reach statistical significance compared to the controls. This is probably due to the low expression of CD16 after long-term culture, which is well described: ADCC effects are generally low in long-term cultured cells due to action of ADAM sheddases, which clip the CD16 molecules and make this process ineffective (Romee et al., 2013). We have included a description of ADCC with proper controls used within the legend of Figure 5 as well as within the Materials and methods section.

4) How do the authors explain the phenotypic and functional differences between the individual ILC1-like subsets as defined by CD5 and CD161 expression and how do these individual subsets contribute to their proposed NKP potential? This point should be discussed in more detail.

We thank the reviewers for the opportunity to share our thoughts of the developmental relationship between the different ILC1-like subsets. We have dedicated a new section within the discussion for this question. We inserted the following into our Discussion:

“The ILC1-like cells could be phenotypically and functionally further broken down into four subsets defined by expression of CD5 and CD161. […] The CD5^+^CD161^+^ subset could constitute an intermediate subset as they possess T cell-specific molecules such as CD2 and chemokine receptors but have already acquire IFNγ effector function.”

5) The authors state that ILC1-like cells preferentially differentiate into mature KIR^+^ NK cells compared to CD56bright NK cells in the OP9-DL1 differentiation setting in the presence of IL-2, IL7 and IL-15. Cytokine stimulation (IL-2 and IL-15) of NK cells leading to the induction of proliferation and results in CD56 and NGK2A upregulation, even in mature CD56dim NK cells. Hence it is not surprising that CD56bright NK cells retained high NKG2A expression while actively proliferating. The present experimental setup therefore does not support their statement in paragraph five of the Discussion of a branched NK cell lineage model.

A problem with the current linear model of NK cell differentiation is that CD56^bright^ NK cells are assumed to be the precursors of CD56^dim^ NK cells but in vitro remain NKG2A^+^ and are very ineffective in developing into more differentiated NK cell subsets, in particular the highly diverse and functionally important NKG2A^-^KIR^+^ subset. This is observed in different in vitro settings such as those using cytokines only but also feeder-supported cultures including murine AFT024, EL-08, OP-9 and even in our human MSC-supported model (Zhao et al., 2018). The unexpected finding in our study was that ILC1-like cells preferentially differentiate into more differentiated, less proliferative NKG2A^-^KIR^+^ NK cells using the same culture conditions as those applied for CD56^bright^. And as discussed above (point 1, second paragraph), the NK cells derived from CD56^bright^ maintain high CD56 expression but in fact differentiate into effector NK cells expressing perforin and GrzB. Thus, the chosen conditions are probably not the reason for the lack of further (terminal) differentiation but might rather be due to the more restricted developmental potential of CD56^bright^ NK cells. We thus discuss a non-linear, “branched" model in which CD56^bright^ NK cell differentiation preferentially leads to NKG2A^+^CD56^dim^ NK cells (which constitute the large majority of CB NK cells) whereas ILC1-like cells preferentially advance to the NKG2A^-^KIR^+^ stage, which would make the roles of CD56^bright^ and ILC1-like cells in building NK cell repertoires complimentary.

6) The authors clearly show that the CD127^+^ Lin- population is highly heterogenous, just by looking at CD161, CD5, CCR9, CCR4 and CCR7. Therefore, the transcriptomic and epigenetic data on the bulk population are not informative. Single cell analysis should be used to define the heterogeneity, considering that Simoni et al. have previously reported the heterogenous nature of human ILC1s (Simoni et al., Immunity, 2017) and questioned the nature of lineages included.

We agree with the reviewer that the transcriptional and phenotypic heterogeneity of circulating ILC1 might go well beyond what was defined in the present study. However, our study demonstrates that the NK cell differentiation potential is present in all major ILC1 subsets defined by single or combined expression of CD161 and CD5. In this regard, we went to great length to show by bulk and single cell cloning experiments that all major subsets similarly possess NK cell differentiation potential characterized by the NKG2A^-^KIR^+^ phenotype and the respective NK cell effector functions. If we further divide these subsets into even smaller populations by single cell RNAseq we will probably identify more subsets but each of these newly defined and very small populations has then to be functionally defined and tested for their individual differentiation potential. We do not think that this kind of analysis would affect the main point of our study that the large majority of circulating ILC1 possess NKP potential.

7) The authors claim that the cells do not generate T cells. However, they only use IL-7 and FLT3, while in other protocols the used IL-7, FLT3L and SCF.

The combination of IL-7 and FLT3L is used in several seminal studies to differentiate CD34^+^ cells into T cells (Schmitt and Zúñiga-Pflücker, 2002; Wang, 2012; Zúñiga-Pflücker, 2004). In addition, the combination seemed to be suitable since ILC1-like cells abundantly express the respective IL-7 receptor (CD127) and the FLT3 ligand. We did not use SCF as this was previously shown to selectively stimulate proliferation of the CD4/CD8 double negative stage while inhibiting further differentiation into the double positive stage in a dose dependent manner (Wang et al., 2006). Nonetheless, we are well aware that there is no consensus on which method to use for a given progenitor and several other approaches are published including classical FTOC and organoid models (Seet et al., 2017). Thus, we put our statement concerning the failure to differentiate into T cells into perspective by stating that although OP9-supported T cell differentiation is widely established it cannot be excluded that the T cell potential might be still present in ILC1-like cells but the in vitro approach was insufficient to deliver all necessary signals present in vivo (subsection “Neonatal ILC1-like cells contain a novel NK cell progenitor”).

[Editors' note: further revisions were suggested prior to acceptance, as described below.]

However, there is still one essential issue which you failed to address to comply with reviewer's recommendations. In our Decision Letter it was explicitly mentioned that "12. The ATAC-sequencing data is based on a single donor. Considering the large variation among individual donors, a sample size of 1 is not adequate to make conclusions". Therefore, unless you address this issues adequately, we cannot be committed to accept your paper for publication in eLife.

We thank the reviewers for pointing this out. We have prepared ATACseq data of ILC1-like cells and NK cells of two additional donors exactly as done before. In accordance with the results of the first sample shown in Figure 4C, the NK cell specific genes CD94/KLRD1, NKG2A/ KLRC1, and KIR2DL3 exhibited high accessibility in NK cells and no accessibility in ILC1-like cells (see Author response image 1). The data reinforce our point that although ILC1-like cells can be readily differentiated into NK cells they are not already poised for transcription of NK cell receptors. We have changed the Legends in the Ms. accordingly. The new files are up-loaded within the same NCBI Project with ID: PRJNA594493.

**Author response image 1. sa2fig1:** Additional ATACseq data. Comparative analysis of regions with open chromatin by ATAC sequencing for KLRD1 (CD94), KLRC1 (NKG2A), and KIR2DL3. For ATAC sequencing, 5000 ILC1-like (left) and NK cells (right) were flow cytometrically sorted to >99% purity. Arrows underneath the ATAC data indicate orientation and start of gene transcription.

**References**

Fink, P.J. 2013. The Biology of Recent Thymic Emigrants. *Annual Review of Immunology* 31:31-50.McFarland, R.D., D.C. Douek, R.A. Koup, and L.J. Picker. 2000. Identification of a human recent thymic emigrant phenotype. *Proceedings of the National Academy of Sciences of the United States of America* 97:4215-4220.

Pekalski, M.L., A.R. García, R.C. Ferreira, D.B. Rainbow, D.J. Smyth, M. Mashar, J. Brady, N. Savinykh, X.C. Dopico, S. Mahmood, S. Duley, H.E. Stevens, N.M. Walker, A.J. Cutler, F. Waldron-Lynch, D.B. Dunger, C. Shannon-Lowe, A.J. Coles, J.L. Jones, C. Wallace, J.A. Todd, and L.S. Wicker. 2017. Neonatal and adult recent thymic emigrants produce IL-8 and express complement receptors CR1 and CR2. *JCI Insight* 2.

Romee, R., B. Foley, T. Lenvik, Y. Wang, B. Zhang, D. Ankarlo, X. Luo, S. Cooley, M. Verneris, B. Walcheck, and J. Miller. 2013. NK cell CD16 surface expression and function is regulated by a disintegrin and metalloprotease-17 (ADAM17). *Blood* 121:3599-3608.

Seet, C.S., C. He, M.T. Bethune, S. Li, B. Chick, E.H. Gschweng, Y. Zhu, K. Kim, D.B. Kohn, D. Baltimore, G.M. Crooks, and A. Montel-Hagen. 2017. Generation of mature T cells from human hematopoietic stem and progenitor cells in artificial thymic organoids. *Nature Methods* 14:521.

Wang, H. 2012. Development of T Cells through Co-culture Lymphoid Progenitor Cells with OP9-DL1 Stromal Cells in vitro. *Bio-protocol* 2:e189.